# Hydrogel-Based Scaffolds: Advancing Bone Regeneration Through Tissue Engineering

**DOI:** 10.3390/gels11030175

**Published:** 2025-02-27

**Authors:** Juan Luis Cota Quintero, Rosalío Ramos-Payán, José Geovanni Romero-Quintana, Alfredo Ayala-Ham, Mercedes Bermúdez, Elsa Maribel Aguilar-Medina

**Affiliations:** 1Faculty of Biology, Autonomous University of Sinaloa, Josefa Ortiz de Domínguez s/n y Avenida de las Américas, Culiacan 80010, Sinaloa, Mexico; luis@uas.edu.mx; 2Faculty of Biological and Chemical Sciences, Autonomous University of Sinaloa, Josefa Ortiz de Domínguez s/n y Avenida de las Américas, Culiacan 80010, Sinaloa, Mexico; rosaliorp@uas.edu.mx (R.R.-P.); geovanniromero@uas.edu.mx (J.G.R.-Q.); 3Faculty of Odontology, Autonomous University of Sinaloa, Josefa Ortiz de Domínguez s/n y Avenida de las Américas, Culiacan 80010, Sinaloa, Mexico; endoalfredo@uas.edu.mx; 4Faculty of Odontology, Autonomous University of Chihuahua, Circuito Universitario Campus I, Chihuahua 31000, Chihuahua, Mexico; mbermudez@uach.mx

**Keywords:** engineering tissue, scaffolds, hydrogel (HG), bone regeneration

## Abstract

Bone tissue engineering has emerged as a promising approach to addressing the limitations of traditional bone grafts for repairing bone defects. This regenerative medicine strategy leverages biomaterials, growth factors, and cells to create a favorable environment for bone regeneration, mimicking the body’s natural healing process. Among the various biomaterials explored, hydrogels (HGs), a class of three-dimensional, hydrophilic polymer networks, have gained significant attention as scaffolds for bone tissue engineering. Thus, this review aimed to investigate the potential of natural and synthetic HGs, and the molecules used for its functionalization, for enhanced bone tissue engineering applications. HGs offer several advantages such as scaffolds, including biocompatibility, biodegradability, tunable mechanical properties, and the ability to encapsulate and deliver bioactive molecules. These properties make them ideal candidates for supporting cell attachment, proliferation, and differentiation, ultimately guiding the formation of new bone tissue. The design and optimization of HG-based scaffolds involve adapting their composition, structure, and mechanical properties to meet the specific requirements of bone regeneration. Current research focuses on incorporating bioactive molecules, such as growth factors and cytokines, into HG scaffolds to further enhance their osteoinductive and osteoconductive properties. Additionally, strategies to improve the mechanical strength and degradation kinetics of HGs are being explored to ensure long-term stability and support for new bone formation. The development of advanced HG-based scaffolds holds great potential for revolutionizing bone tissue engineering and providing effective treatment options for patients with bone defects.

## 1. Introduction

Tissue engineering is a multidisciplinary biomedical field focused on developing technologies and methodologies to promote tissue renovation, replacement, and regeneration [1,2]. The primary goal of tissue engineering is to create biological substitutes using cellular and extracellular components to restore the functionality of injured or impaired organs and tissues [3]. This emerging area of research holds vast potential and diverse biomedical applications, mostly due to the incorporation of HGs. These biomaterials possess a highly flexible structure resembling the extracellular matrix (ECM), making them suitable for use as grafts [4]. Additionally, the integration of bioactive materials in tissue engineering facilitates bone repair by filling defects and accelerating the healing of damaged tissues [5,6].

Biomedical techniques for tissue replacement traditionally rely on autologous grafts, which, despite their advantages, are associated with certain drawbacks, such as the risk of infection [7]. The selection of the most appropriate material for such procedures is contingent upon several factors, including tissue viability, as well as the size, shape, and volume of the defect [8]. In response to these challenges, regenerative medicine emerges as an innovative approach by integrating tissue engineering with the body’s intrinsic self-healing capabilities to treat tissues that are otherwise incapable of repairing themselves [9,10]. Central to this approach are biomaterials, which serve as adaptable platforms for tissue engineering applications, tailored based on their biocompatibility, degradation kinetics, and mechanical properties [11]. Biomaterial-based scaffolds are categorized into bio-polymers, bio-metals, bio-ceramics, and bio-composites [12,13]. Another classification considers their consistency: for instance, certain modifiable materials exhibit strong compression and tensile resistance-like bone, yet are hydrophobic, which limits cell encapsulation, and their low degradation rate delays effective healing. For optimal outcomes, biomaterials must degrade at a rate proportional to new bone formation [14].

Bone regenerative therapies have earned significant attention in recent years. These therapies require the synergistic use of three critical components: (1) scaffolds such as HGs made from biomaterials to provide structural support, (2) growth factors to stimulate cellular activity, and (3) local cell populations to facilitate tissue regeneration. This triad is designed to mimic natural biological processes that promote fracture healing or the repair of critical-sized defects, ultimately restoring damaged biological functions; furthermore, the incorporation of exogenous cells can enhance the osteogenic potential of these therapies [15,16].

Three key features are also essential: the capability of a matrix to retain and transport cells to a specific site, to support the repair process by native cells capable of forming a functional matrix, and to promote the synthesis of bioactive molecules such as cytokines and growth factors of the host, as well as to transport molecules added in the HG to promote bone repair [17]. In other words, these scaffolds are employed to facilitate cellular adhesion and the subsequent formation of new tissues, providing an ideal environment for cell adhesion and proliferation; this environment often mimics the ECM within a highly porous three-dimensional structure [12,18]. Such biomaterials offer additional advantages, including the possibility of administration via minimally invasive surgical procedures or as injectable systems, involving self-setting materials that solidify under physiological conditions or soft gels designed to adapt to the defect site, thereby improving their clinical applicability and patient outcomes [19,20]. In this context, HG-based scaffolds, with a viscous, gelatin-like consistency, offer excellent degradation rates and permeability, allowing the efficient exchange of oxygen, nutrients, and metabolites, despite their mechanical properties [14,21]. Natural polymers such as chitosan, collagen, agarose, and alginate are the most commonly used in tissue engineering [21], and stand out due to their unique properties and high biocompatibility [22].

Synthetic polymer-based scaffolds, such as polylactic acid (PLA), are extensively studied for their applications in load-bearing implants, including fracture fixation devices, due to their biodegradable nature and favorable mechanical strength [23]. Also, polyethylene glycol (PEG) is valued for its excellent solubility in water and organic solvents, making it an effective modifier for starch/PLA composites, as it combines the biodegradability capacity of starch with the toughness of PLA [24]. Furthermore, polycaprolactone (PCL), an FDA-approved biocompatible and bioresorbable polyester, has shown potential in medical applications. Incorporating PCL into brittle PLA matrices enhances their toughness, creating a polymeric matrix more suitable for bone-related applications [25]. All these synthetic polymers stand out for their adjustable physicochemical properties, enabling the replication of physiological environments. Thus, adaptability highlights the complementary roles of synthetic and natural polymers in advancing tissue engineering applications for bone regeneration. Thus, this review aimed to investigate the potential of natural and synthetic HGs, and the molecules used for its functionalization, for enhanced bone tissue engineering applications.

## 2. Bone: A Highly Specialized and Dynamic Tissue

Bone, among other characteristics, is a dense, calcified, and porous connective tissue [26], and one of the hardest tissues in the human body, just below the enamel of the teeth [27]. The human skeleton comprises cortical and trabecular bone, which differ in structure despite identical chemical composition. Cortical bone, comprising 80% of the skeleton, is dense, compact, and slow-renewing, providing mechanical strength. Trabecular bone, accounting for 20% of bone mass, is less dense, more elastic, and metabolically active, with a higher turnover rate [28].

Bone is a mineralized, dense connective tissue primarily composed of a mineral component and an organic matrix that mainly consists of type I collagen (COL-I), with smaller amounts of type III collagen (COL-III). Hydroxyapatite, represented as Ca_3_(PO_4_)_2_·(OH)_2_, forms nanocrystals that are embedded between individual collagen molecules, significantly increasing the rigidity of bone. In addition to collagen, non-collagenous proteins such as fibronectin, osteocalcin (OCN), osteopontin (OPN), and osteonectin contribute to a scaffold for mineral deposition [29]. Bone is a highly specialized and dynamic tissue that undergoes constant regeneration. The process of bone remodeling is regulated and coordinated by various cell types. Maintaining bone remodeling and overall mineral homeostasis relies on a delicate balance between bone resorption and formation [30]. The bone remodeling cycle is defined by five stages: activation of osteoclast precursors, resorption carried out by differentiated osteoclasts, the reversal phase lasting approximately 4 to 5 weeks, the bone formation phase regulated by osteoblasts, and finally, the termination phase, where osteoblasts differentiate into osteocytes [31].

### Bone Healing

Bone possesses a remarkable ability to repair itself following both, microtraumas and macrostructural injuries, such as fractures. This repair process is mediated through neurogenesis, angiogenesis, osteogenic differentiation, and biomineralization [32]. Bone healing is a naturally occurring phenomenon that can be categorized into two distinct mechanisms: primary and secondary repair. Primary repair occurs when the fracture gap is smaller than 0.1 mm, enabling direct ossification across the gap without the formation of an intermediary callus. In contrast, secondary repair, which is the predominant mechanism, takes place when the fracture gap exceeds 0.1 mm but is less than twice the diameter of the injured bone. This process involves a multistage sequence of biological events, including the formation of a callus that eventually undergoes ossification [33].

The physiological mechanisms underlying bone repair can further be divided into two primary pathways: direct and indirect healing. Direct healing, also known as intramembranous ossification, is characterized by the differentiation of mesenchymal progenitor cells directly into osteoblasts, resulting in the immediate deposition of bone matrix. Indirect healing, in contrast, is mediated by an intermediate phase of cartilage formation, known as endochondral ossification. This process relies on the differentiation of chondrocytes, which play a pivotal role in forming a cartilage template that is subsequently replaced by mineralized bone [34].

Bone repair is orchestrated by a highly complex cascade of cellular and molecular events involving various cytokines (e.g., IL-1, IL-6, TNF-α) and growth factors. These bioactive molecules are essential for the recruitment of progenitor and inflammatory cells, including lymphocytes, macrophages, eosinophils, and neutrophils, to the site of injury. These cells coordinate the initial inflammatory response, establish the regenerative microenvironment, and initiate downstream signaling pathways critical for tissue regeneration and repair (Figure 1) [34]. The repair of bone is a meticulously regulated process that unfolds in distinct sequential phases, including hematoma formation, an acute inflammatory response, the development of granulation tissue, bone regeneration, and subsequent remodeling [35].

For successful bone repair, the injury must not exceed a critical size; otherwise, natural recovery through the immune system becomes unlikely, requiring external interventions such as bone grafting to expedite the healing process [36,37]. Approximately 5% of the millions of skeletal fractures that occur each year result in nonunion, where the bone defect fails to heal effectively with the initial treatment and remains mechanically unstable [38]. According to the Global Burden of Disease organization, 178 million new bone fractures were reported worldwide in 2019, representing a 33.4% increase since 1990. Lower limb fractures, including those of the patella, tibia, fibula, and ankle, were the most common, with an age-standardized rate of 419.9 cases per 100,000 population [39]. In addition, degenerative bone diseases affect more than 1.7 billion people worldwide; thus, bone is one of the most frequently transplanted tissues worldwide, with more than 4 million medical procedures performed annually to address bone-related conditions [40]. Furthermore, bone void-filling treatments serve more than half a million patients annually, representing a market valued at USD 2.8 billion [41].

In the United States, more than 450,000 patients are affected by bone and cartilage defects annually, requiring numerous bone grafting procedures and a substantial number of autologous transplants each year [10,16], with autologous bone grafts, particularly those from the iliac crest [9], due to their three key regenerative properties: osteoconductive, osteoinductive, and osteogenic capacity [42,43].

## 3. Hydrogel-Based Scaffolds and Their Role in Tissue Engineering

HG-based scaffolds have been extensively used across various domains, including agriculture, pharmaceuticals, cosmetics, and biomedicine, with significant relevance in tissue engineering applications [44]. The development of tissues in artificial regenerative environments requires scaffolds that replicate the microstructure of the ECM, thereby facilitating the repair of damaged or diseased tissues and organs [26,45]. Both synthetic and natural HGs present a viable and less invasive alternative, addressing the limitations inherent to conventional scaffolds [1].

These HGs provide a three-dimensional architecture that supports tissue regeneration, establishing them as a cornerstone in regenerative therapies. This structure creates an optimal environment for the proliferation and differentiation of endogenous cells while ensuring effective nutrient exchange and waste removal [46]. Moreover, the structural stability of HGs is achieved through various cross-linking methods, including physical and chemical approaches (Table 1), which also enable the controlled release of bioactive agents, such as drugs and growth factors, encapsulated within their matrix [47]. Critical factors influencing physical cross-linking include temperature, pH, and the electrostatic interactions inherent to the polymeric components, highlighting their importance in the functional optimization of these materials [47].

Regarding osteoinductivity, the use of growth factors is essential during the synthesis of various HGs. In addition to growth factors, peptides such as parathyroid hormone-related peptide (PTHrP), arginine–glycine–aspartic acid (RGD), and LL37 are also used. However, bone morphogenetic proteins (BMPs) have proven to possess the greatest osteoinductive potential. These growth factors significantly increase the expression of Runx-2 and osterix (OSX), mediating both the Smad and MAPK pathways. It is crucial that during the synthesis of HGs, they possess the ability to maintain and sustain the controlled release of these growth factors and peptides, thereby contributing to tissue regeneration and repair [48]. A key factor is the method of growth factor binding to the HG. Covalent binding of the BMP-2 growth factor via amino groups has been shown to enhance cell adhesion and differentiation in culture. However, this synthesis method is both time-consuming and costly. Alternatively, non-covalent interactions between the HG and the growth factor such as electrostatic interactions, hydrophobic forces, hydrogen bonds, and van der Waals forces facilitate a simpler binding approach but present the disadvantage of an uncontrolled initial release, which may lead to ectopic bone formation [49].

**Table 1 gels-11-00175-t001:** Types of HGs for scaffolds design.

Physical-Responsive HG		Chemical-Responsive HG	
Temperature-responsive HG	Characterized by the presence of hydrophobic groups in the monomers such as methyl, ethyl, and propyl	pH-responsive HG	They work especially in the administration and incorporation of bioactive agents.
Photo/light-responsive HG	They contain a photoreceptor fraction, where photoirradiation induces a reaction involving isomerization (cis–trans, open–close), cleavage, and dimerization, which is then transferred to the functional group	Glucose-responsive HG	Developing insulin delivery system. They promote water absorption capacity and an increase in the size of the matrix.
Electro and magnetic-responsive HG	They change their properties in response to small changes in electrical current stimuli or external fields	Biological/biochemical-responsive HG	They contain biological functionalities/instructive fragments designed to interact with the biological environment. They use biomolecules such as peptide sequences that act as cross-linkers.

[50].

## 4. Key Issues in HG Selection

One of the key factors for achieving satisfactory outcomes in tissue engineering, specifically in bone regeneration, is the selection of the HG. In recent years, the use of aggregated MSCs has gained attention due to their pluripotent properties, self-renewal capacity, and multispectral differentiation potential [51]. Some contraindications of using mesenchymal stem cells (MSCs) include the potential formation of heterotopic bone, particularly in joints, which can impair their normal function. Therefore, it is crucial to regulate homeostasis and control the regenerative process through strategies that prevent ectopic bone formation. The use of HGs with nanoparticles containing curcumin-loaded zeolitic imidazolate and cerium oxide has been shown to modulate immune activity, promoting inflammatory balance and enhancing efferocytosis, thereby reducing the likelihood of heterotopic bone regeneration [52].

Other materials used to regulate immune functions during bone regeneration include the incorporation of peptides such as MP196, which is rich in arginine and tryptophan, into HGs. These peptides have been shown to upregulate the expression of genes associated with neutrophil apoptosis, macrophage polarization towards the M2 phenotype, and the promotion of efferocytosis, thereby contributing to the establishment of a favorable microenvironment for bone regeneration [53].

Another important factor in choosing HGs is their capacity to cause minimal or no inflammation. As mentioned earlier, macrophage polarization to M1 and M2 phenotypes plays a critical role in bone regeneration. In vitro studies using bone marrow-derived macrophages (BMDMs) have demonstrated that calcium alginate/amelogenin-based HGs promote macrophage polarization towards the M2 phenotype, as evidenced by the expression of CD206. These findings also show an inhibition of the inflammatory response collaborating in regenerative processes [54].

Finally, an important aspect in material selection is that it should not trigger a foreign body reaction, such as fibrotic encapsulation. While commonly used materials, such as polyethylene glycol (PEG), exhibit excellent biocompatibility, studies have shown that they can still induce fibrotic encapsulation. Research on HGs with alternating sequences of glutamic acid and lysine has reported minimal inflammation two weeks after subcutaneous implantation, as well as no evident collagen encapsulation after six months in mice. These findings open new possibilities for the use of such materials in the field of tissue engineering [55].

## 5. Physical Properties of Hydrogels

### 5.1. Cross-Linking

One way that HGs are classified is based on their cross-linking method, which can be physical through non-covalent cross-links or chemical through intra- or interpolymeric covalent bonds. Some studies on gelatin-based HGs report that chemical cross-linking enhances the network strength of the hydrogel and increases its water absorption capacity. Fourier transform infrared (FTIR) spectroscopy evaluations show a significant increase in the strength of triple helix structures, which resemble collagen [56]. The cross-linking method plays a crucial role in modifying the physical structure of HGs. Some materials that serve as cross-linking agents while also enhancing the physical properties of HGs include resveratrol and tannic acid. In the case of polyvinyl alcohol (PVA), it lacks the ability to form hydrogels independently at temperatures of 25 °C or higher. Therefore, it requires the addition of cross-linking agents such as resveratrol, which facilitates HG formation through strong hydrogen bonds between both materials, as confirmed by attenuated total reflection FTIR spectroscopy [57].

### 5.2. Stiffness

Stiffness refers to the ability of a material to withstand forces or stress, measured by the elastic modulus or Young’s modulus. This is a critical property of HGs, particularly those intended for load-bearing sites, as it depends on polymer molecular weight and cross-linking density. For MSC-based applications, the stiffness of HGs ranges from 0.1 kPa to over 100 kPa [58]. This property plays a key role in MSC differentiation, as these cells sense hydrogel stiffness by exerting forces on the substrate to which they adhere [59].

The combination of different polymers significantly influences HG stiffness. A formulation consisting of gelatin/polyethylene glycol diacrylate/2-(dimethylamino)ethyl methacrylate demonstrated stiffness values exceeding 170 kPa and compressive strength above 700 kPa, making it a promising material for bone regeneration [60].

The impact of stiffness is further supported by studies on N-carboxyethyl chitosan hydrogels mixed with oxidized hyaluronic acid at different concentrations (5%, 7.5%, and 10% wt). The resulting stiffness values were 576.36 Pa, 3842.662 Pa, and 7368.937 Pa, respectively. Notably, higher stiffness was correlated with greater cell proliferation rates of BMSCs at days 1, 7, and 14, highlighting its importance in cellular responses [61].

### 5.3. Porosity

Another important characteristic of HGs is their porosity, as it plays a crucial role in cell proliferation and migration. The porous structure of HGs influences nutrient diffusion, waste removal, and cellular interactions, all of which are essential for effective tissue regeneration. In humans, the porosity of both trabecular and cortical bone varies between 5% and 90%, while the pore diameter ranges approximately from 0.1 to 2000 µm [62]. Research reports that HGs composed of calcium carbonate microcapsules, nanohydroxyapatite, chitosan, and collagen with varying hydroxyapatite concentrations (0%, 1%, 2%, and 5%) exhibit good porosity, with pore diameters below 100 µm when synthesized with 2% nanohydroxyapatite. However, their mechanical properties, including strength, viscoelasticity, and physical stability, decrease compared to HGs with higher nanohydroxyapatite concentrations [63].

### 5.4. Water Absorption

Another property is the water absorption capacity of HGs, which is a key property for promoting cell adhesion and migration, essential for tissue regeneration. This feature is often evaluated using the contact angle test to assess swelling behavior. Hydrogels composed of chitosan/β-glycerophosphate sodium/polyvinyl alcohol with varying antler powder deer concentrations (6%, 8%, and 10%) exhibit different contact angles, with the 10% formulation showing the most favorable results. However, no direct correlation has been established with water absorption capacity, as the 6% formulation shows no significant difference from the others [64].

## 6. Bioactive Properties of Hydrogels

Finally, it is important to highlight the bioactive properties that HGs should possess, such as controlled drug or metal ion (e.g., Cu) release, antibacterial activity, reactive oxygen species (ROS) scavenging, inflammation inhibition, and promotion of M2 macrophage polarization. These properties contribute to the establishment of a microenvironment that supports osteoblast and endothelial cell adhesion, proliferation, migration, and differentiation while minimizing osteoclast recruitment [65].

### 6.1. Drug Delivery

Bone tissue damage is not always solely the result of trauma; underlying diseases can also impede bone regeneration under homeostatic conditions. Therefore, HGs designed for bone regeneration must possess the capacity to be loaded with pharmacological agents that actively promote the regenerative process [66].

HGs composed of GeIMA and chitosan loaded with acetylsalicylic acid (aspirin) have demonstrated satisfactory drug release capabilities in promoting bone regeneration. Viability and proliferation assays conducted on adipose tissue-derived stromal cells (ADSCs) after 24 h of exposure to the HGs showed high proliferation levels. Additionally, the long-term sustained release of aspirin contributes to anti-inflammatory effects and enhances osteoinductive properties [67].

### 6.2. Osteogenic Potential

Various HGs exhibit biocompatibility, degradability, and low cytotoxicity, among other properties. However, their osteogenic potential is often weak or even absent. Therefore, it is necessary to incorporate bioactive components or biological factors into HGs to enhance or maximize their osteogenic capabilities.

One example is the use of amine-modified bioactive glass (BAG) nanoparticles in polyethylene glycol (PEG) HG, which exhibit strong osteogenic properties, as evidenced by the upregulation of osteogenesis-related genes such as RUNX2, ALP, OCN, and OSTERIX in bone marrow-derived mesenchymal stem cell (BMSC) cultures [68].

On the other hand, the use of osteocyte cultures, which have been less explored, has demonstrated remarkable effects on the osteogenic potential of HG. The interstitial fluid velocity within the bone matrix and vertical channels ranges between 0.04–0.2 and 0.06–0.9 m/s, respectively. By modifying the interstitial fluid flow velocity in MLO-Y4 osteocyte cultures, it is possible to stimulate anabolic responses in osteocytes, leading to a downregulation of osteoclastogenesis-related genes and the production of soluble mediators that induce osteoblastic differentiation, ultimately contributing to new bone formation [69].

### 6.3. Proliferation Potential

Another crucial bioactive property of HGs is their ability to promote the proliferation of resident cells at the injury site as well as the cells incorporated into the HG, particularly mesenchymal stem cells (MSCs). Bioactive molecules such as peptides are commonly used to enhance this proliferative capacity. The E7 peptide, a short peptide with the amino acid sequence EPLQLKM, has been shown to influence MSC proliferation. In gelatin methacrylate Wharton’s jelly hydrogels functionalized with E7, results indicate that cell proliferation within E7-modified HGs is significantly higher compared to control groups, confirming that this peptide can effectively enhance proliferative properties [70].

Another material that enhances the proliferative bioactivity of HGs is graphene oxide. In vitro experiments involving the encapsulation of single MSCs within alginate micro-hydrogels containing graphene oxide nanolayers have demonstrated the ability to maintain MSC viability and proliferation, as assessed through live/dead assays and fluorescence microscopy. These findings highlight the potential application of graphene oxide-functionalized HGs in minimally invasive tissue regeneration therapies [71].

## 7. Hydrogel-Based Scaffolds for Bone Regeneration

One of the primary characteristics required for HGs is biocompatibility, i.e., the ability of a material to avoid provoking a harmful immune response in the body [47], while performing its intended function, such as supporting bone regeneration [21]. HGs require structural and design properties that protect active molecules, ensure nutrient permeability, and mimic the native bone environment. Their porous structures are essential for nutrient uptake, waste diffusion, and intercellular communication (Figure 2) [17,72]. These hydrophilic polymeric products [73], with a great capacity to absorb and retain significant amounts of water [74], form a three-dimensional structural network due to the weak interactions between the polymeric chains [75]. This network is stabilized by cross-linking within the polymer structure, which may involve covalent bonds, ionic bonds, and physical interactions such as hydrogen bonding, van der Waals forces, and physical entanglements [21,74]. Typically, these HGs contain between 1% and 20% dry mass [15] and can be directly transplanted to the injury site after being molded into a specific form in vitro through controlled cross-linking [16].

According to Calcei and Rodeo, osteoconductive, osteoinductive, and osteogenesis are the three major properties of orthobiologics used in bone healing [42]. In efforts to achieve bone regeneration using natural or synthetic materials, the principles of osteoconductivity, osteoinductivity, and osteogenesis, are fundamental. Osteoconductive refers to the use of scaffolds to passively support the growth of cells, tissues, and blood vessels. Osteoinductive involves the application of growth factors that induce the differentiation of MSCs, while osteogenic capacity denotes the presence of cellular elements within a material that enables it to generate new bone independently [42,43].

In this regard, HGs could achieve these three properties depending on the form of use: Osteoconductive HGs: hydrogels based on ECM have molecules such as proteoglycans, collagen, and/or gelatin, have molecules such as arginine–glycine–aspartic acid (RGD), and have a variety of low-molecular-weight peptides that give them properties that favor the regeneration of bone tissue without the need to be enriched with any other molecule, activating the migrations of resident cells of the tissue. Osteoinductive HGs: in addition to having their regenerative properties, they are enriched or functionalized with molecules that are not part of the HG itself, such as growth factors that amplify the therapeutic activation of resident cells by exogenous factors to generate new bone. Osteogenic HGs: these HGs are enriched with osteoprogenitor cells capable of generating new bones at the site of the injury and the capacity to elicit bone formation in ectopic sites (Figure 3).

### 7.1. Natural Polymers

The most commonly used polymers in tissue engineering are derived from natural sources formed via photosynthesis or biochemical reactions in a biological environment [21]. These include materials such as collagen, alginate, starch, cellulose, gelatin, fibrin, silk, hyaluronic acid, and chitosan (Figure 4) [76].

Natural polymers exhibit excellent biocompatibility with minimal immunogenicity. They facilitate cell adhesion and proliferation, support tissue regeneration, and contribute to the mechanical stability and structural integrity of bone. However, their primary limitations include low mechanical strength and susceptibility to rapid degradation in specific biological environments, influenced by enzymatic activity and implantation site (Table 2) [23].

#### 7.1.1. Protein-Based Natural Polymers

##### Collagen

Collagen, as a flexible, cationic polymer, is the most abundant protein in the extracellular matrix [104]. Its main functions are related to providing mechanical support to cells, tissues, and blood vessels, as well as offering resistance to compressive forces [105]. It is sourced from various animal species, with well-established protocols for its isolation and purification involving enzymatic or chemical treatments [106]. Collagen possesses a range of physicochemical properties, including biocompatibility, biodegradability, and cross-linking potential, due to the presence of various functional groups, such as hydroxyl, amino, carboxyl, guanidyl, and imidazole groups. These groups confer unique physical and chemical characteristics, making collagen highly suitable as a compound in tissue engineering applications [21].

COL-I is the most utilized form in tissue engineering and HG formulation. Additionally, its high concentration of RGD ligands facilitates material–cell interactions and adhesion [107]. Is primarily secreted by osteoblasts and undergoes mineralization in the later stages of bone development, providing an optimal microenvironment for tissue engineering applications [16]. In vitro studies have demonstrated that collagen micro- and nanostructured scaffolds possess mechanical compression properties that are approximately 50% comparable to human trabecular bone, without compromising porosity. Furthermore, viability assays—such as those assessing adhesion, proliferation, and metabolic activity, including alkaline phosphatase (ALP) activity and mineralized matrix synthesis—have yielded promising results when compared to lactic acid and PEG scaffolds using human osteoblasts (HOBs) [3]. These findings highlight the superior performance of collagen-based HGs over other types for bone regeneration [3]. It has also been demonstrated that the combination of implanting dense collagen HGs containing wild-type murine dental pulp stem cells (mDPSCs) with weekly systemic injections of a sclerostin antibody (Scl-Ab) significantly enhances bone regeneration in critical-sized calvarial defects created in mice [108]. There is also evidence of the development of collagen-based HG scaffolds infused with tacrolimus, which were surrounded by a PCL/gelatin membrane, the scaffolds were characterized for their potential use in bone tissue engineering applications, and the in vivo study provided preliminary evidence of the efficacy of these HGs in treating bone defects [109].

This biomaterial exhibits properties that promote the proliferation of urine-derived stem cells (USCs) in both 2D and 3D models. The HG consisted of a composite of chitosan microspheres loaded with BMP-2 embedded within a type I collagen HG that released growth factor for over 28 days, enhancing ALP and promoting calcium mineral deposition in 2D models. In vivo assays demonstrated a significant volume of newly formed bone eight weeks after HG administration in cranial defects in Sprague Dawley rats [110].

A significant drawback of collagen HGs as scaffolds for tissue engineering is their variability, influenced by factors like collagen source and gelation pH, which can result in difficulties during the synthesis of the material. Additionally, collagen’s rapid degradation, weak mechanical strength, opacity, and high shrinkage limit its use in tissues that need more stiffness, such as bone or cartilage [107,111].

##### Fibrin

Fibrin is a commonly utilized scaffolding material in tissue engineering and regenerative medicine [112,113]; notably, fibrin exhibits minimal inflammatory responses and demonstrates excellent biocompatibility, making it a promising candidate for biomedical applications [78]. Is found within blood clots, and it has the unique capacity to carry bone morphogenetic proteins (BMPs); notably, fibrin can undergo enzymatic cross-linking, forming a gel with adhesive properties that facilitate injectable application [16]; however, its application is constrained by a lack of responsiveness to external stimuli in situ, inadequate mechanical properties, pronounced shrinkage, and rapid biodegradation, which limit its suitability for cell culture [112,113].

A straightforward method for fabricating fibrin HGs involves combining fibrinogen and thrombin at 37 °C, which also enables the modification of the gel’s internal fiber structure and porosity [76]. In vitro and in vivo studies of fibrin HGs combined with BMP-2 (37.5 µg/mL) have demonstrated osteogenic properties. In vitro assays revealed that sustained BMP-2 release after 28 days, along with increased ALP expression compared to control groups and fibrin HGs without BMP-2. In vivo, statistically significant increases in total bone volume were observed at tibial defect sites in Wistar rats treated with the BMP-2-enriched fibrin HG, compared to those treated with fibrin HGs alone [77].

Another study reported the mechanical properties and biodegradability of a fibrin HG combined with konjac glucomannan. In vitro assays demonstrated biocompatibility and the ability to enhance osteogenesis of nasal mucosa-derived ectodermal mesenchymal stem cells (EMSCs) by upregulating mineralization and ALP. In vivo results further showed the improved reconstruction of alveolar bone defects, as evaluated by micro-computed tomography (µCT) in Sprague Dawley rat models [114].

##### Gelatin

Gelatin is a material that, due to its ability to be used as a drug delivery system for osteogenic active molecules and scaffolds, is widely used in tissue engineering. It is a natural polymer obtained by the hydrolysis of collagen obtained from pig skin and bone, and bovine skin and bone [115].

These materials offer physicochemical advantages, such as a naturally occurring negative charge that interacts with positively charged inductive molecules added to them, a property that can be adjusted during the manufacturing process [36]. In this context, these HGs have also been combined with molecules such as oxidized chondroitin sulfate and mesoporous bioactive glass nanoparticles, capable of cross-linking in situ under physiological conditions in the presence of borax. This combination improves compressive strength, enhances the differentiation of bone marrow-derived mesenchymal stem cells (BMSCs) into osteoblasts, and successfully facilitates the regeneration of cranial bone defects in rats [116].

The chemical coupling of polydopamine-modified gelatin HGs with polyetheretherketone using glutaraldehyde solutions has demonstrated sustained release capabilities of bone BMP-2. Tests on MSC cultures also demonstrated an increase in run-related transcription factor 2 (RUNX2) expression associated with increased BMP-2. In addition, the production of OCN, ALP, and calcium deposition increased due to the incorporation of the growth factor, proving that this HG maintains a potential for bone regeneration [117].

This material also offers promising results in decreasing the risk of delayed bone consolidation when loaded with a black phosphorus nanosheet and deferoxamine. In vitro studies showed the overexpression of RUNX2, COL-I, and BMP-4 and the proliferation of BMSCs. In vivo results in Sprague Dawley rats demonstrated satisfactory results 4 weeks after the administration of the material on 3 mm diameter defects in tibial bone evaluated macroscopically and histologically [118].

Other combinations of this HG have similarly demonstrated preosteoblast differentiation and high cell viability when biphasic calcium phosphate is incorporated into gelatin HGs [119]. Studies are reporting the ability of gelatin-based HGs to be cross-linked using visible-wavelength light when combined with riboflavin as a photoinitiator. This approach achieves a stiff range that supports the proper differentiation of osteoblasts and the proliferation of KUSA-A1 cells [120].

##### Extracellular Matrix (ECM)

Natural scaffolds derived from the ECM are widely utilized in preclinical research and clinical trials [26]. For bone regeneration, they promote the development of new bone and maintain homeostasis within a complex environment of biochemical and biomechanical signals that collectively regulate pluripotency, proliferation, and differentiation of resident cells [84]. Additionally, they can be combined with other materials or polymers, such as collagen and chitosan [81]. It has been reported that the combination of bone ECM (bECM) with dental pulp stem cells enhances cell viability after three weeks in culture and promotes the expression of sialoproteins (BSPs) and RUNX-2, even in the absence of osteogenic inducers [121].

The ECM obtained from human bone tissue has emerged as a promising candidate in regenerative medicine. By decreasing the particle size of bone powder and extending enzymatic digestion times, it is possible to achieve higher protein concentrations and diversity, alongside improved gelation properties. Bone powder with particle sizes ranging from 45 to 250 µm, tested on MSCs for 7 days, significantly promoted osteogenic differentiation, evidenced by the upregulation of key genes such as ALP, RUNX2, COLA1, and OCN. This approach represents a valuable addition to the strategies for bone regeneration [122].

The ECM derived from porcine dermis loaded with biphasic calcium phosphate powder has also been utilized for bone regeneration. It has demonstrated the capability to be administered as an injectable material due to its thermosensitive properties and favorable mechanical characteristics. MC3T3-E1 pre-osteoblasts showed no evidence of cytotoxicity and exhibited enhanced osteogenic differentiation through the overexpression of genes such as bone BSPs, OPN, and ALP, as analyzed by quantitative real-time PCR. Additionally, in vivo assays on defects in the femoral head of rabbits demonstrated new bone formation at 4 and 8 weeks, with superior outcomes observed in HGs containing 15% biphasic calcium phosphate [123].

##### Silk and Silk Fibroin

Silks possess a distinctive combination of properties including biocompatibility, biodegradability, low immunogenicity, and ease of accessibility, and can be obtained from natural sources, such as silkworms [103]. Biomineralized materials from silk spider proteins are widely used in bone regeneration [124]. For instance, the combination of MaSp1 protein and bone BSP in the development of scaffolds promotes the adhesion of MSCs and contributes to their differentiation [125].

The gelation process can be achieved via various methods, including physical cross-linking techniques such as self-assembly, shear force application, ultrasonication, and electric field exposure. Chemical cross-linking methods, including photopolymerization, radiation exposure, and enzymatic cross-linking, are also employed for silk [78].

However, silk fibroin exhibits limited osteoconductive properties; therefore, researchers have incorporated various bioactive molecules to address this limitation [126]. When combined with Laponite (LAP), silk fibroin demonstrates the ability to regenerate bone in calvarial defects in rat models and elicits an osteogenic response in BMSC cultures [126].

The combination of silk fibroin and fibrin with amorphous calcium phosphate (ACP) and platelet-rich plasma (PRP) has been applied to the repair of 5 mm cranial bone defects in Sprague Dawley rats. New bone formation was observed 8 weeks later and evaluated through µCT. Additionally, the seedbed HGs demonstrated succinate-like activity, marked by the upregulation of solute carrier family 13 member 3 (SLC13A3), which subsequently activated the PI3K/Akt signaling pathway. These findings highlight its potential as a biomaterial with significant bone regeneration capabilities for addressing critical-sized defects [127].

In the same context, silk fibroin combined with calcium phosphate particles and methylcellulose enhance the biological properties of the material while also providing the opportunity for the HGs to be administered in an injectable form, gelating at a physiological temperature in approximately 40 min. An osteoblast precursor derived from *mus musculus* calvarial mouse cells (MCT3T3-E1) exhibits low cytotoxicity, with viability exceeding 98% under concentrations of 10% and 15% *w*/*v* of calcium phosphate particles. Furthermore, this material also demonstrates the ability to promote osteogenic differentiation of the cell line after 7 days [128].

#### 7.1.2. Carbohydrate-Based Natural Polymer

##### Alginate

Alginate is the most widely used biomolecule of marine origin, just below cellulose [129], with good biocompatibility and biodegradability, and has been used for cell delivery or as a protein carrier in bone tissue engineering applications [87]. Extracted primarily from brown algae (*Phaeophyceae*) using alkaline solutions such as calcium chloride (CaCl_2_) [21], alginate is a natural anionic biopolymer composed of alternating α-L-guluronic acid (G) and β-D-mannuronic acid (M) residues [44]. This polymer is capable of absorbing large amounts of biological fluids, making it effective in regulating homeostatic and wound-healing processes [21].

In vivo assays in rats have demonstrated the ability of these HGs, loaded with allogeneic BMSCs, to regenerate bone defects mimicking cleft lip and palate injuries. The regeneration was evaluated through µCT and histological sections at different time points [87].

In addition to combining these HGs with MSCs, the adhesive properties of alginate have been enhanced by conjugating it with other molecules, such as adhesion-promoting ligands like RGD [130].

This polymer can be combined with various materials, such as polyvinyl alcohol (PVA) and bioactive glass (BAG), demonstrating excellent biocompatibility in hemolysis and cytotoxicity assays using MC3T3-E1 cells. The fabrication of this HG also enhances ALP activity, and the expression of genes associated with osteogenic differentiation, indicating its potential for bone repair applications [131].

Finally, alginate methacrylate HGs have been developed as bioinks for bioprinting scaffolds, providing alternative approaches in manufacturing. These HGs have shown promising results in the expression of genes associated with osteogenesis [89].

##### Hyaluronic Acid

Hyaluronic acid is a naturally occurring glycosaminoglycan (polysaccharide) abundant in soft connective tissues, such as the skin, cartilage extracellular matrix, and synovial fluid [79]. With its low toxicity, hypoallergenic nature, biocompatibility, and biodegradability, hyaluronic acid is widely used in biomedicine for applications like scaffold manufacturing in tissue engineering, wound healing, and drug delivery systems [132]. As a non-sulfated glycosaminoglycan, hyaluronic acid significantly impacts properties within the ECM like cell motility mediation, and other capabilities of the material such as lubrication and water retention. Additionally, it plays a role in cell interaction with growth factors and osmotic pressure regulation [17].

In combination with niobium-doped BAG, it can promote new bone formation through the RUNX2 signaling pathway [92]. These HGs, in combination with hydroxyapatite and short peptides such as fluorenylmethoxycarbonyl-diphenylalanine, have demonstrated favorable outcomes in promoting the osteogenic differentiation of MC3T3-E1 preosteoblasts. Additionally, they facilitate calcium deposition, as evidenced by Alizarin Red staining [93].

Studies report the use of hyaluronic acid and PVA-Gel that can be rapidly administered via injection, forming through cross-linking of its multifunctional derivatives. When enriched with BMP-2, this material has demonstrated the capability to generate ectopic bone in muscle tissue, observed through computed axial tomography at 4 and 10 weeks post-injection. In vitro MTT assays on human dermal fibroblasts (HDFa) revealed no restrictions in cell proliferation, highlighting the material’s promising osteogenic potential for bone regeneration [133]. They are also capable of supporting exosomes while preserving their biological activity and enabling sustained release at bone defect sites [134]. In addition, it can blend with synthetic polymers such as PEG and PCL to form HGs loaded with solid lipid nanoparticles, demonstrating satisfactory biocompatibility on BMSCs evaluated through live/dead staining. Furthermore, this HG combination exhibits a modulatory capacity on macrophage polarization between M1 and M2 phenotypes, reducing M1 polarization while enhancing M2 polarization, thereby contributing to bone regeneration. This effect has been confirmed through evaluations of critical cranial defects in rats using µCT analysis [135].

##### Chitosan

Chitosan ranks as the second most abundant biodegradable polymer after cellulose, commonly extracted from the exoskeletons of insects, fungal mycelia, and crustaceans [27,95]. Structurally, it consists of D-glucosamine and N-acetyl-D-glucosamine units linked by β-1,4-glycosidic bonds, with a molecular weight range from 10 to over 1000 kDa [27,44,79,136]. It is a natural polycationic polymer with hydrophilic properties, derived from the partial deacetylation (40% to 80%) of chitin [17,137]. Its unique attributes make it suitable for biomedical applications, including biocompatibility, biodegradability, non-toxicity, and strong mucoadhesive properties, as well as a wide spectrum of antibacterial activity.

Chitosan exhibits oxygen permeability and a structure closely resembling glycosaminoglycans, which also enhances its biological functionality. This biopolymer is widely applied in areas such as hemostasis, wound healing, transdermal patches, skin and bone regeneration, cartilage tissue engineering, blood vessel embolization, drug delivery systems, and gene expression modulation [17,27,138]. Chitosan HGs can be fabricated through physical or chemical methods. Chemical cross-linking involves agents, irradiation, and secondary polymerization, while physical cross-linking relies on electrostatic, hydrophobic, and hydrogen-bonding interactions [21]. Aqueous chitosan solutions at physiological pH values or above 6.2 precipitate due to ionic forces, making them challenging to handle. However, combining chitosan with glycerol phosphate enables gelation at physiological temperatures, which improves its usability in surgical applications [17].

Chitosan is also adaptable for injectable applications and can be combined with growth factors, such as BMPs, to enhance its regenerative potential. Luca et al. conducted in vivo and in vitro trials combining chitosan HGs with BMP-2 and β-tricalcium phosphate; this composite showed promising results: in murine models, it successfully induced ectopic bone formation, and in rabbit femur models, it contributed to lesion regeneration. Although the regeneration was incomplete, this may have been due to the limited duration of the study or the critical size of the bone defect, which is often challenging for natural healing processes [139].

Chitosan has also been combined with injectable rat-tail collagen and supplemented with extracellular vesicles derived from osteoblasts to promote bone regeneration. This approach yielded favorable results, particularly in enhancing the ECM mineralization of BMSCs compared to gels without vesicles [94]. Satisfactory results have also been observed in alveolar bone regeneration by incorporating basic fibroblast growth factor (bFGF) and transforming growth factor β3 (TGFβ3). Chitosan microspheres containing these growth factors have demonstrated the ability to regenerate alveolar bone defects in Sprague Dawley rats while ensuring sustained long-term release of TGFβ3 and 95% release of bFGF within the first 7 days. In vivo evaluations using µCT have shown bone regeneration, facilitated by the nutrients assimilated by stem cells due to microvascular formation promoted by bFGF [140]. Finally, it has been shown that chitosan upregulates OPN and OCN gene expression in human amnion-derived stem cells (HAMSCs) from amniotic tissue, as confirmed by RT-PCR assays after 21 days [81].

##### Dextran

Dextran is a biodegradable and biocompatible exopolysaccharide synthesized by lactic bacteria [141]. This biopolymer is composed of glucose monomers with an α-1,6 bond [142]. This neutral glucan, composed of glucose monomers with numerous active hydroxyl groups, allows for easy chemical modification, enhancing its utility in biomedical applications; it is biodegradable and exhibits exceptional solubility, biocompatibility, and non-immunogenic properties [21]. To enhance the bioactivity of the material, various strategies can be employed. One effective approach involves incorporating inorganic compounds, such as hydroxyapatite, β-tricalcium phosphate, or BAG, into the polymeric matrix [100]. When combined with PEG and gelatin, these HGs enhance the colonization of human BMSCs. Furthermore, the inclusion of osteogenic factors promotes mineralization after 28 days [98]. This combination of dextran with gelatin has also demonstrated the manipulation of pore size and distribution, a key feature in regenerative materials. When this combination is used with bioprinting, it promotes the proliferation, migration, and expansion of BMSCs while activating signaling pathways associated with Yes-Associated Protein (YAP) [143].

The capacity of dextran to generate apatite crystals is null; however, when combined with BAG, the formation of crystals has been demonstrated, and it shows favorable cell viability, as evaluated through MTT assays in human osteosarcoma cell cultures (SaOS-2) [99]. Other studies have demonstrated the potential of dextran microspheres incorporated within hyaluronic acid scaffolds. These experiments revealed promising cell viability when human adipose-derived stem cells (hASCs) were seeded onto the scaffold, with fluorescence microscopy indicating favorable live/dead cell ratios [96]. Furthermore, when photobiomodulation was applied, hASCs exhibited enhanced differentiation characteristics, with increased ALP and calcium deposition observed within 24 h [144].

In vivo experiments in C57BL/6J mice with femur defects treated with dextran–tyramine conjugate HGs showed satisfactory results, evaluating regeneration with µCT when the HG was loaded with basic fibroblastic growth factor (bFGF) [145]. Similar results have been observed in cranial defects in rats when dextran HGs are loaded with human umbilical vein endothelial cells (HUVECs). However, when this material is loaded with both HUVECs and human osteoblasts in co-culture, the outcomes are less favorable. Alternatively, the use of stromal-derived growth factor (SDF) presents a viable option for improving regenerative outcomes [146].

### 7.2. Synthetic Polymers

Synthetic HGs can control their physicochemical properties through adjustments during their synthesis. They are materials composed of synthetic polymer networks and can be mass-produced, which surpasses natural materials, making them specific for different scenarios (Table 3) [147].

They exhibit lower bioactivity and biocompatibility compared to natural polymers; however, they can be enhanced during manufacturing processes to enable drug and growth factor loading, thereby increasing their regenerative potential [148].

These materials play a crucial role in the field of tissue engineering, often being used in conjunction with growth factors and short peptides derived from proteins naturally found in tissues to support cell viability and the direct phenotype of proliferation to enhance their therapeutic potential [149]. While they exhibit little or no osteoconductive properties, their effectiveness in achieving regenerative outcomes is often limited when used alone; they typically require the incorporation of additional molecules or bioactive materials [76].

Synthetic polymers’ versatility facilitates the modification of their physical properties, molecular weights, functional groups, configurations, and chain conformations. This adaptability enables the production of various shapes with high reproducibility. Furthermore, their exceptional tensile strength and resistance make them capable of providing mechanical support while simultaneously degrading over time [150,151].

**Table 3 gels-11-00175-t003:** Synthetic polymer scaffolds; properties and function.

Synthetic Polymer Scaffolds	Properties	Applications	Scaffold Synthesis	References
Polyethylene glycol (PEG)	Injectable Cytocompatibility Increase collagen deposition	Bone regeneration Dental pulp regeneration Cell and drug carrier	Photopolymerized cross-linking	[35,152,153,154]
Polypropylene fumarate (PPF)	Biocompatibility Biodegradability Viscous (21 °C) Injectable Elasticity Hardness	Bone regeneration Substitute for trabecular bone Drug carrier	Covalent bond cross-linking	[155,156,157,158]
Polycaprolactone (PCL)	Biocompatibility Biodegradability Osteogenic properties	Bone regeneration Periodontal regeneration	Covalent bond cross-linking	[13,159,160,161,162]
Polyvinyl alcohol (PVA)	Biocompatibility Cytocompatibility Mechanical stability high water absorption capacity	Bone regeneration	Physically cross-linking	[57,163,164]
Poly (γ-glutamic acid) (γ-PGA)	Biocompatibility Non-toxicity Tailorable degradation Encapsulation characteristic	Bone regeneration	Glucose cross-linking	[165,166,167]

#### 7.2.1. Polyethylene Glycol (PEG)

Poly(ethylene glycol) (PEG) is extensively utilized in the fabrication of well-defined HG due to its exceptional biocompatibility and flexibility. However, its application is hindered by limitations such as low mechanical strength and inadequate cell adhesion. These drawbacks can be effectively addressed through engineering strategies, such as the integration of cell adhesion motifs and matrix metalloproteinase (MMP)-sensitive cross-links, which facilitate cellular infiltration and matrix remodeling in both in vitro and in vivo environments [168].

When combined with polylactic acid (PLA), the surface hydrophilicity is enhanced, as evidenced by a reduction in the water contact angle. Additionally, its degradability increases upon exposure to phosphate-buffered saline (PBS) solutions over 8 weeks. Furthermore, the material demonstrates low cytotoxicity when tested in cells that has fibroblast morphology isolated from the bone patient with osteosarcoma (MG-63) cell cultures [152]. When combined with BAG, the material demonstrates enhanced osteogenic properties along with significant improvements in mechanical characteristics, including compressive strength, flexibility, and extensibility [153]. Furthermore, PEG HGs have been utilized for cell encapsulation; however, they may elicit foreign body responses. Studies involving murine macrophages (RAW 264.7) and MC3T3-E1 preosteoblasts encapsulated in PEG HGs exposed to lipopolysaccharide (LPS) have shown that macrophage stimulation leads to a 5.3-fold increase in apoptotic osteoblasts, a 4.2-fold reduction in ALP, and a 7-fold decrease in collagen deposition [154].

This compound exhibits excellent biocompatibility, as demonstrated by viability assays such as MTT and live/dead staining in human vascular endothelial cells (HUVECs) and BMSCs. When combined with poly(L-lysine) and dendrimers, it significantly enhances the expression of BMP-2, indicating its osteogenic potential [35]. In addition to enhancing BMP-2 expression, this polymer can encapsulate the growth factor to further support bone formation. The incorporation of RGD as a functional group promotes cell adhesion, particularly of osteoblasts, resulting in effective bone regeneration. This has been demonstrated in cranial defects in rats, with µCT evaluations at 4 weeks post-administration confirming significant bone regeneration [168].

#### 7.2.2. Polypropylene Fumarate (PPF)

PPF is a material with medical applications due to its biological properties such as biodegradability. It has a fumaric acid-based structure with the main disadvantage of not having osteoinductive or osteoconductive properties [169]. As a derivative of fumaric acid, PPF exhibits a viscous consistency at room temperature (21 °C) and is characterized by its cross-linking mechanism, which occurs via covalent bonds. This property facilitates its manipulation in conjunction with other cross-linking agents, enabling in situ cross-linking upon injection [170]. Additionally, reports have highlighted its antibacterial capacity when antibiotics such as rifampicin and ciprofloxacin are incorporated. In vitro infection models demonstrate a reduction in bacterial load, while simultaneously supporting the formation of new bone tissue [156].

Further studies report that PPF effectively encapsulates PGA microspheres for the controlled release of BMP-2 growth factors. This system has been evaluated in bone stromal cell cultures (W20-17) by assessing ALP and drug release profiles. In in vivo bone defect in rats, PPF’s regenerative capacity has been validated through µCT imaging, confirming its osteogenic effectiveness [158].

When PPF is combined with methoxy PEG and enriched with flavonoid nanoparticles, it improves the expression of osteogenesis-related genes, including COL-I, OCN, and OPN. This effect is observed 14 days after the in vitro exposure of MG-63 cells to the HG. These in vitro results position the material as a promising candidate for applications in bone regeneration [171].

#### 7.2.3. Polycaprolactone (PCL)

PCL is a polymer with high hydrophobicity as its main disadvantage; however, it is characterized by its high biocompatibility, design flexibility, low degradation rate, high stability, and cost-effectiveness, making it an ideal candidate for tissue engineering applications, particularly in periodontal and bone tissue regeneration. Its nanofibers have been widely employed as scaffolding materials for effective tissue repair and regeneration processes [159]. It has shown the ability to regenerate critical bone defects when combined with zinc after 8 weeks, as well as activating the Wnt/β-catenin and NF-κB signaling pathways, as verified in in vitro models using MC3T3-E1 and RAW264.7 cells [159].

Scaffolds based on PCL enriched with magnesium microparticles (3% weight) enhance osteogenic properties [13]. These scaffolds promote the expression of key genes involved in bone metabolism, such as OPN, OCN, Col1a1, and RUNX2, while also demonstrating biomineralization through Alizarin Red staining [162].

The functionality of PCL in controlled molecule release has been demonstrated, particularly with growth factors such as BMP-2. Studies have shown that PCL enables a complete and sustained release of BMP-2 over 21 days, as corroborated by in vitro assays utilizing MSCs, highlighting its potential in tissue regeneration applications [160].

#### 7.2.4. Polyvinyl Alcohol (PVA)

Polyvinyl alcohol (PVA) is widely utilized due to its high hydrophilicity, biocompatibility, low toxicity, and chemical stability. It is a polymer derived from the hydrolysis of polyvinyl acetate [57]. When combined with natural molecules capable of self-assembling with PVA, supramolecular hydrogels can be formed, resulting in enhanced mechanical properties [163].

PVA-based hydrogels incorporating resveratrol have demonstrated favorable viscoelastic mechanical properties, as assessed through rheological tests. Additionally, in vitro studies report that hydrogels containing 0.4% resveratrol effectively promote alkaline phosphatase (ALP) activity and mineral deposition. Furthermore, this HG upregulates genes associated with osteogenic activity, including BMP-9, OCN, and ALP in MC3T3-E1 cells, positioning it as a promising material in the field of bone regeneration [57].

Other studies support the favorable mechanical properties of PVA-based hydrogels at various concentrations, particularly when combined with chitosan, which enhances mechanical performance. Notably, a 2% PVA concentration has been shown to provide high water absorption capacity.

Furthermore, the degradation behavior of these hydrogels in phosphate-buffered saline (PBS) containing 100 mg/L of lysozyme has yielded promising results. Finally, cytotoxicity assays using MG63 cells demonstrate a statistically significant increase in cell proliferation compared to a positive control (DMSO), further supporting the biocompatibility and potential biomedical applications of these materials [164].

#### 7.2.5. Poly(γ-glutamic acid) (γ-PGA)

A water-soluble polyamino acid composed of glutamic acid units linked by peptide bonds through amino and carboxyl groups [165]. It is primarily produced by Gram-positive bacteria of the *Bacillus* species [166].

The combination of γ-PGA with whey protein isolate HGs has demonstrated a high water absorption capacity before degradation begins. Additionally, cytotoxicity assessments at days 3 and 14 indicated minimal toxicity. Regarding its osteogenic potential, evaluations using MC3T3-E1 cells showed significantly higher alkaline phosphatase (ALP) activity compared to the control group, suggesting its suitability for in vivo studies [166].

Collagen-based HGs containing γ-PGA loaded with BMP-2 have yielded promising results in bone regeneration, with evaluations conducted both in vitro and in vivo. These HGs exhibit low viscosity at room temperature, transitioning to high elastic and viscous moduli under physiological conditions. Furthermore, they support BMMSC proliferation, attributed to their high porosity and upregulation of ALP, BSP, and OCN at days 7 and 14. Finally, in vivo assays in calvarial defect models of mice demonstrated complete recovery within 4 weeks, as assessed by micro-CT (µCT) [167].

## 8. Biomodels, Type of Bone Defect and Tests for the Evaluation of Bone Defect Regeneration

For the in vivo evaluation of HGs in bone regeneration studies, the literature reports promising results across various materials. Regarding biomodels, the most commonly used are rodents, specifically Sprague Dawley rats, Wistar rats, C57BL/6J mice, and Fisher 344Jcl rats. Several factors contribute to the preference for these models, including their genetic similarity to humans, estimated at approximately 99%. Additionally, their handling in laboratory animal facilities is significantly easier compared to larger biomodels such as dogs or primates (Table 4).

Rodents also offer cost-effectiveness, as they require minimal maintenance, consume relatively little food, and are easy to manage. Furthermore, their rapid reproductive cycle facilitates the establishment of large populations in a short period, and they reach sexual maturity quickly, making them highly suitable for in vivo studies.

## 9. Functionalization of HGs Using Growth Factors and Peptides for Bone Regeneration

Bone regeneration is driven by a complex cascade of growth factors [178]. Scaffolds and HGs must incorporate chemical signals, including cell adhesion sites, growth factors, and other signaling molecules, to improve the bioactivity of the scaffold and support the maintenance of homeostasis [179]. Bioactive molecules like growth factors and peptides play a pivotal role in bone tissue engineering by enhancing the functionality of biomaterials to facilitate bone remodeling on cranial defects in rats [172]. Additionally, various components and factors significantly impact the effectiveness of HG, including the incorporation of MSCs, the bioactive molecules used for their functionalization or combination, and the selection of the most suitable cell lines for in vivo evaluations (Table 5).

### 9.1. Growth Factors

Bone defects can be effectively regenerated through the application of scaffolds enriched with appropriate growth factors (frequently BMPs) [16,36,176]. This is attributed to the electrostatic interactions between the positively charged growth factors and the negatively charged HG, which facilitate the physical immobilization of the factors within the scaffold [36]. Growth factors serve as crucial agents in regulating cellular behavior and enhancing the functionality of newly formed bone tissues by promoting angiogenesis, thereby ensuring an adequate supply of oxygen and nutrients [15,36]. These factors play pivotal roles in chondrogenesis, osteogenesis, and vascularization and have demonstrated promising outcomes when integrated into HGs [15]. Conversely, the use of growth factors administered without HGs, typically through direct injections at the target site, has been associated with limitations such as reduced bioavailability, a short half-life, and adverse effects, including the formation of ectopic spindles and significant inflammation in surrounding tissues [78].

#### 9.1.1. Bone Morphogenic Protein (BMP)

BMPs function as stimulatory agents for the differentiation of MSCs into various cell types, including chondrocytes, osteoblasts, fibroblasts, myocytes, and adipocytes [15,34,43]. This makes them a viable therapeutic option for enhancing local bone regeneration [158,160,172].

BMP-2 has been shown to yield positive outcomes in osteogenesis and the healing of spinal injuries; however, its use is contraindicated in patients with a history of cancer, as it has been associated with the promotion of cancer cell proliferation [9]. Clinical trials involving BMP-2-enriched HGs have demonstrated favorable outcomes in bone formation within nasopalatine clefts at a concentration of 250 µg/mL, particularly when compared to iliac crest autografts, as assessed via µCT six months post-grafting [180].

When BMP-2 is encapsulated in chitosan and PEG HGs, it ensures a sustained release profile while promoting bone regeneration in cranial defects in rats. Additionally, in vitro assays using BMSCs have demonstrated that HGs containing this growth factor enhance the expression of osteogenic differentiation-related genes, such as RUNX2, OPN, and OCN. This activity facilitates osteoblast differentiation, bone formation, and development, while also contributing to the regulation of bone metabolism, highlighting the therapeutic potential of BMP-2 functionalized HGs in bone tissue engineering [174].

In vitro assays have demonstrated the potential of bone morphogenetic protein 9 (BMP-9) as a potent inducer of osteogenic differentiation in MG63 cell cultures. Notably, BMP-9 also enhances its expression, amplifying its osteogenic effects. Furthermore, in vivo studies conducted on femoral bone defects in rabbits corroborate these findings, showing satisfactory bone regeneration. This outcome was assessed at 1, 4, 12, and 24 weeks post-treatment using µCT, highlighting the consistent and robust regenerative capabilities of BMP-9 in bone tissue engineering applications [181].

#### 9.1.2. Platelet-Derived Growth Factor (PDGF)

PDGF is an FDA-approved therapy specifically utilized to promote bone regeneration [178]. Platelet is a potent source of growth factors like PDGF that can be extracted through centrifugation methods from blood treated with anticoagulants [79]. Both in vitro and in vivo studies demonstrate the efficacy of platelets and PDGF in promoting tissue regeneration. In a study by Chen et al., an HG enriched with salts such as calcium and sodium, combined with platelets, was utilized for in vitro analysis with rat-derived MSCs and in vivo analysis in Sprague Dawley rats. Viability assays revealed significant increases in cell count and proliferation within the HG compared to controls at 48 h. Additionally, enhanced alkaline phosphatase activity and expression of osteogenic markers such as OCN, OPN, and RUNX2 were observed. The in vivo findings further indicated statistically significant improvements in tissue density, thickness, and bioavailability of the HG [19].

Studies suggest that the use of PDGF positively impacts the regeneration of gaps induced in primary osteoblast cell cultures and MC3T3-E1 cells, promoting junction formation through the upregulation of connexin via the activation of p-Akt signaling, thereby highlighting its regenerative potential [182].

PDGF has also been utilized in microspheres of nanohydroxyapatite incorporated into gelatin methacryloyl (GelMA) scaffolds. The results demonstrated that it not only induced the recruitment of MSCs derived from rabbit bone marrow but also promoted the osteogenic differentiation of these stem cells, as confirmed through cell migration assays, ALP, and Alizarin Red staining [183].

Finally, the potential of PDGF is further corroborated through the use of biomimetic periosteum-bone scaffolds, where electrospun poly-L-lactic acid (PLLA) fibers are deposited on the surface of a gelatin/chitosan cryogel, with platelet-derived growth factor-BB (PDGF-BB) encapsulated within the core of PLLA nanofibers in a core–shell structure. The PDGF-loaded scaffold exerts a synergistic effect on osteogenesis, significantly accelerating bone healing. In vitro experiments demonstrated that the biomimetic periosteum-bone scaffolds exhibit favorable biocompatibility and osteogenic capacity. Additionally, in vivo experiments showed that the composite scaffold effectively and rapidly repaired rat cranial defects [5].

#### 9.1.3. Fibroblast Growth Factor (FGF)

FGF are a class of proteins fundamental to the regulation of various cellular processes and tissue development [79]. In vivo studies in monkeys have demonstrated promising outcomes for cranial defect repair within 21 days post-implantation [177]. A comparable effect is observed when bFGF is used in dental pulp stem cells and silver-loaded titanium implants, demonstrating osteogenic differentiation. Furthermore, bFGF exhibits anti-inflammatory properties by inhibiting M1 macrophage activity [184]. In the context of alveolar bone regeneration and height augmentation in Wistar rats, FGF has demonstrated promising results in addressing bone defects. Moreover, FGF effectively inhibits epithelial tissue overgrowth in the affected area, thereby creating a favorable environment for bone formation. These outcomes position FGF as an ideal growth factor for periodontal tissue regeneration, where bone plays a critical role in restoring structural and functional integrity [173].

Although FGF does not directly participate in signaling pathways that promote bone regeneration, its role is crucial in initiating the healing process of such defects. When combined with BMP-2 in gelatin methacrylate HG, FGF significantly enhances bone regeneration outcomes. Additionally, FGF effectively stimulates the proliferation of BMSCs, a finding supported by robust in vitro assays, further highlighting its potential in bone tissue engineering applications [185]. Additionally, in vivo studies on mandibular defects in rats, evaluated using µCT, have demonstrated that the use of resorbable collagen membranes loaded with 0.5 µg and 2 µg of FGF-2 effectively promotes bone defect regeneration. The release of the growth factor is sustained for up to six days before gradually declining, highlighting its controlled release profile and its potential to facilitate bone tissue repair [175].

### 9.2. Peptides

Functional peptides have demonstrated efficacy in promoting bone regeneration and macrophage polarization to M1 and M2 phenotypes, an important point since M1 macrophages are involved in the beginnings of bone healing, promoting the inflammation and infection control necessary for these processes, and finally, M2 is involved in regeneration itself. However, their rapid degradation, physicochemical instability, and limited in vivo half-life remain significant limitations [186].

#### 9.2.1. Arginine–Glycine–Aspartic Acid (RGD)

The primary ligand studied, RGD, is a peptide that binds to integrins. The covalent binding of RGD to alginate scaffolds significantly enhances bone formation. However, high concentrations of adhesive ligands, while promoting adhesion, can impede cellular migration; thus, optimizing adhesive properties is essential to maintain a balance that supports both adhesion and migration [15]. RGD peptides exhibit an affinity for integrins expressed by osteoblasts. Studies on dental implants loaded with these peptides report higher levels of newly formed bone tissue, suggesting their contribution to improved implant stability [187]. RGD peptides are frequently incorporated into scaffolds to enhance cell adhesion. The results demonstrate that the addition of RGD peptides to titanium alloy, PCL, and phosphate complexes significantly promotes bone repair and healing [188].

#### 9.2.2. LL37

Cathelicidin, also known as LL-37, is a 37-amino-acid peptide with a cationic charge, extensively involved in the innate immune response, exhibiting both pro-inflammatory and anti-inflammatory activities depending on the specific tissue context [189]. However, LL-37 is associated with several limitations, including high production costs, reduced activity under physiological conditions, susceptibility to proteolytic degradation, and significant cytotoxicity in human cells [190].

The sequential release of peptide W9 following LL-37 release has been shown to further enhance the osteogenic differentiation of BMSCs, demonstrating a synergistic effect in promoting bone tissue regeneration [191]. Studies on cranial defects in rats have provided evidence of LL-37’s efficacy in bone regeneration. During the initial phases of tissue repair, fibroblast-like cells have been observed, indicative of early regenerative activity. Moreover, LL-37 is associated with the recruitment of stem cells, as demonstrated by the expression of the STRO-1 gene, further supporting its role in bone tissue engineering applications [192]. Other works have also shown that the peptide promotes the proliferation and migration of BMSCs. Regarding osteogenic differentiation, the presence of the peptide (10 µg/mL) promotes the expression of related genes such as ALP and OCN [193].

Studies utilizing 3D scaffolds composed of silk fibroin, chitosan, nanohydroxyapatite, and the peptide LL-37 have demonstrated a reduction in bacterial load and subsequent bone regeneration in murine models. These findings are supported by in vivo RNA sequencing analyses that reveal gene expression related to bone remodeling, as well as assessments of ALP and calcium mineral staining with Alizarin Red [194].

#### 9.2.3. Osteogenic Growth Factor Peptide (OGP)

Osteogenic growth factor peptide (OGP) is a tetradecapeptide [195] with a highly conserved 14-amino-acid motif [196]. In vitro studies indicate that OGP-enriched gelatin HGs can upregulate the expression of BMP-2, OCN, and OPN genes, as well as facilitate calcium salt deposition by murine MCT3T-E1 pre-osteoblasts [197].

## 10. Difficulties and Adverse Effects of HGs and Their Components

A variety of products for tissue engineering are approved by the FDA and EMA, underscoring the importance of HGs in this field [198]. Numerous studies emphasize the limitations and complications of current clinical methods for bone repair, including autografts and allografts [73]. Adverse effects, such as ectopic bone formation in muscle tissue, have been reported with BMP-2-enriched HG. Studies show that incorporating bioceramic agents like hydroxyapatite into gelatin-based HGs reduces the required BMP-2 concentration from 2–15 µg to 600 ng, facilitating a more controlled release profile [198]. Challenges also include the need for HGs to maintain the stability of incorporated molecules, such as growth factors or peptides, which rely on cross-linking structures with precisely tuned degradation properties to ensure controlled, sustained drug release [138]. Synthetic or non-biological HGs often provoke a localized innate immune response, beginning with nonspecific protein adsorption, followed by macrophage persistence, chronic inflammation, and eventual fibrous encapsulation of the material [154].

Additionally, the degradation properties of HGs play a crucial role in achieving effective bone regeneration. Ideally, the degradation rate should be perfectly synchronized with new bone formation. Materials such as gelatin methacryloyl GelMA HGs combined with BAG have shown promising results in degradation assays using collagenase-containing media within the first 24 h, significantly enhancing the functional properties of HGs [199]. Other materials that contribute to the degradation rate of HGs, in addition to BAG, include chitin whiskers. After 14 days, the porosity of the HG increases, becoming more elongated and partially collapsed. When exposed to enzymatic solutions for 21 days, the degradation rate reaches 19.78 ± 1.49%, supporting cellular differentiation and in vitro osteogenesis [200].

Another challenge associated with HGs is their mechanical properties.

A clear example of the limitations of hydrogels, particularly injectable formulations, is their inability to regenerate bone tissue in load-bearing sites due to their lack of compressive strength [201].

For instance, while gelatin methacryloyl (GelMA) HGs offer advantages in tissue engineering—such as cell culturing, drug delivery, and tissue regeneration—due to their similarity to the extracellular matrix (ECM), they exhibit low mechanical strength. This limitation can be addressed by incorporating materials like magnesium phosphate cement (MPC), which has demonstrated promising results in enhancing the viscoelastic properties of HGs. Studies evaluating different MPC concentrations (2.5%, 5%, and 7.5%) have shown significant improvements in mechanical performance [202].

## 11. Conclusions

HG-based scaffolds represent a transformative advancement in the field of bone tissue engineering, offering a synergistic combination of biocompatibility, biodegradability, and tunable properties. By mimicking the ECM and providing a supportive environment for cellular adhesion, proliferation, and differentiation, these scaffolds facilitate the regeneration of bone tissue. The incorporation of natural polymers, such as collagen and chitosan, and synthetic polymers, like PEG and PCL, enhances the mechanical and bioactive characteristics of these materials, optimizing their performance in regenerative therapies. Furthermore, the functionalization of HGs with bioactive molecules, such as growth factors and peptides, amplifies their osteoconductive and osteoinductive capacities, driving superior outcomes in bone repair and regeneration. Despite their promise, ongoing research is essential to address challenges such as mechanical limitations and variability in degradation rates, ensuring the long-term efficacy of these innovative biomaterials. This approach holds significant potential for addressing the growing global demand for effective and minimally invasive treatments for bone defects.

## 12. Future Perspective

Tissue engineering has achieved remarkable progress in regenerative therapies, evolving from the use of natural materials capable of inducing bone growth to synthetic materials with tunable mechanical properties and reduced risks of rejection. Despite these advancements, significant challenges remain in bridging the gap between experimental outcomes and clinical trials.

While the mechanical properties of synthetic materials have improved considerably, critical fractures in long bones often necessitate rigid fixation devices to ensure stability. Although minimally invasive surgical techniques have advanced significantly in accelerating bone regeneration, certain cases still require pre-formed scaffolds specifically designed to match the topographical and anatomical characteristics of the defect.

The integration of bioprinting and bio-ink technologies, incorporating cells and growth factors [203], represents a key area of focus in tissue regeneration research within the biomedical sciences. These innovative approaches offer the potential to provide the structural rigidity necessary to emulate the properties of bone [204] while simultaneously reducing the dependence on expensive fixation materials. As a result, these strategies hold significant promise, warranting further development to enhance treatments for bone degeneration and improve patient outcomes. When used with 3D bioprinting technology in combination with stimuli-responsive bioinks, this approach is referred to as 4D bioprinting [205].

Finally, it is important to highlight smart HGs, which are designed to respond to external stimuli such as temperature, pH, electric and magnetic fields, ionic strength, and enzymatic environments. Their key characteristics include controlled drug delivery, biomedical applications, biosensors, and their use in tissue engineering and regenerative medicine [50].

## Figures and Tables

**Figure 1 gels-11-00175-f001:**
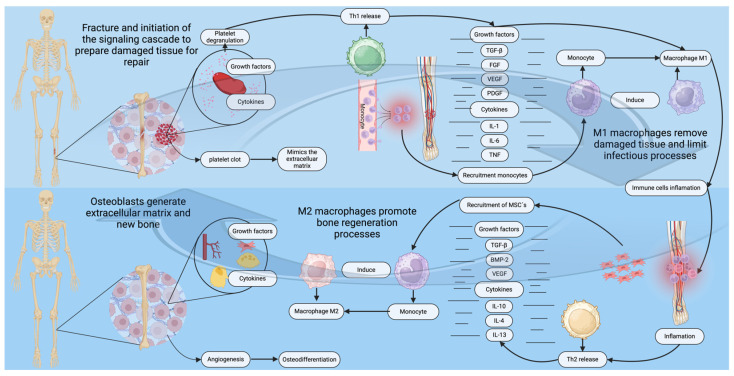
Fracture repair is a process that involves several physiological phenomena in a specific order: fracture of the periosteum and bone, immediately after which the formation of a clot begins to imitate the environment of a matrix, then the onset of inflammation thanks to platelet degranulation that releases growth factors like vascular endothelial growth factor (VEGF), fibroblast growth factor (FGF), platelet-derived growth factor (PDGF), and transforming growth factor β (TGF-β), and then cytokines like interleukin 1 (IL1), interleukin 6 (IL6), and tumor necrosis factor (TNF) induce macrophage polarization into the M1 phenotype. The expansion and proliferation of mesenchymal stem cells (MSCs) are initiated when macrophages undergo M2 polarization and release tissue repair signals, including interleukin 10 (IL-10), transforming growth factor-beta (TGF-β), bone morphogenetic protein-2 (BMP-2), and vascular endothelial growth factor (VEGF), thereby promoting angiogenesis. It is important to note that the application of MSCs has been extensively studied in vitro, in vivo using animal biomodels, and in clinical trials. The rationale for MSC utilization lies in their dual role: they can autonomously differentiate into bone cells (autocrine function) while also enhancing bone regeneration and repair through the secretion of various growth factors (paracrine function). Under these conditions, the process of bone formation (endochondral and intramembranous) begins, with fibroblasts penetrating the defect area to build a fibrous callus, and finally the recruitment of osteoclasts that will eliminate the primary bone and remodel the bone structure.

**Figure 2 gels-11-00175-f002:**
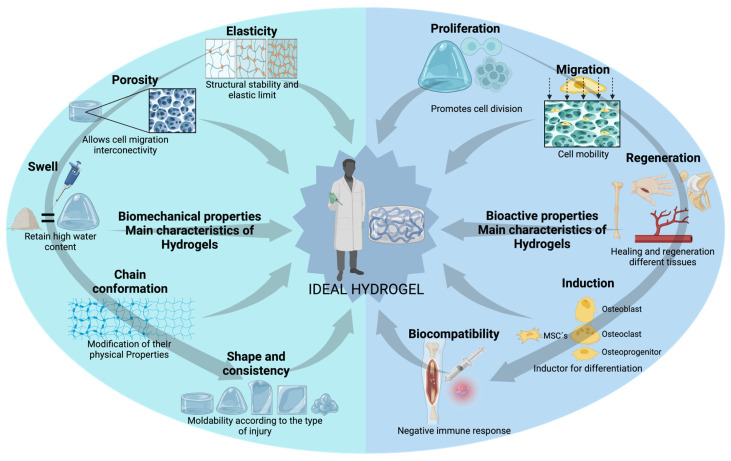
There are various properties and characteristics that make HGs ideal materials for the regeneration of different tissues, such as bone. These properties can be classified into biomechanical and bioactive categories. Biomechanical properties include attributes that facilitate the administration of HGs at the injury site, such as their shape and consistency relative to the conformation of their polymer chains, ensuring minimal procedural complexity. Additionally, their pore size and high water absorption capacity promote the retention and migration of exogenous agents incorporated into the HG, thereby supporting the regenerative process. Regarding bioactive properties, HGs exhibit high biocompatibility due to their natural or synthetic origins. Depending on the manufacturing process and their combination with cells and growth factors, these materials can enhance and improve their inductive and proliferative capabilities, ultimately leading to the successful regeneration of damaged tissues.

**Figure 3 gels-11-00175-f003:**
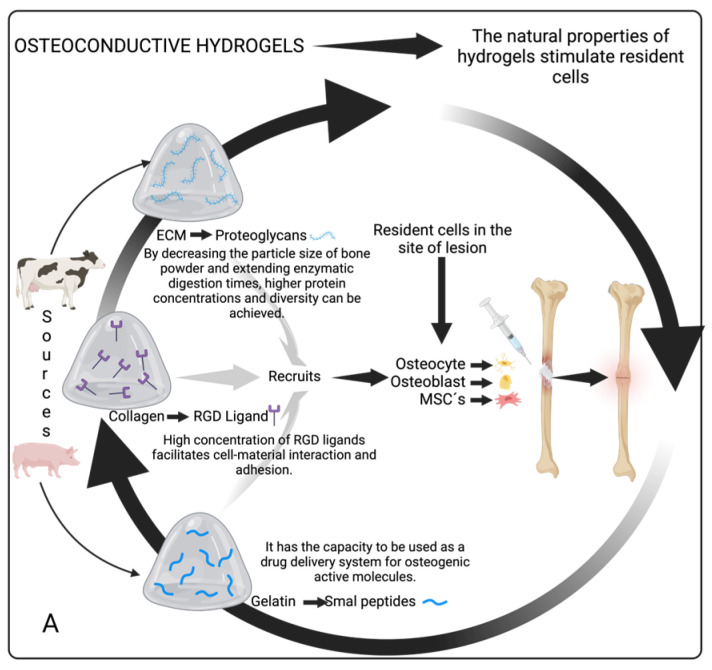
HGs could be classified according to their regenerative capacities. (**A**) Osteoconductive HGs: hydrogels based on ECM, collagen, and/or gelatin have their molecules such as proteoglycans, RGD, and a variety of low-molecular-weight peptides that give them properties that favor the regeneration of bone tissue without the need to be enriched with any other molecule, activating the resident cells of the tissue, and promoting cell migration to the injury site. (**B**) Osteoinductive HGs: in addition to having their regenerative properties, they are enriched or functionalized with molecules that are not part of the HG itself, such as growth factors that amplify the therapeutic action of the scaffold by inducing the resident cells to generate new bone by exogenous factors. (**C**) Osteogenic HGs: these HGs are characterized by being enriched by osteoprogenitor cells capable of generating new bone at the site of the injury or ectopic sites.

**Figure 4 gels-11-00175-f004:**
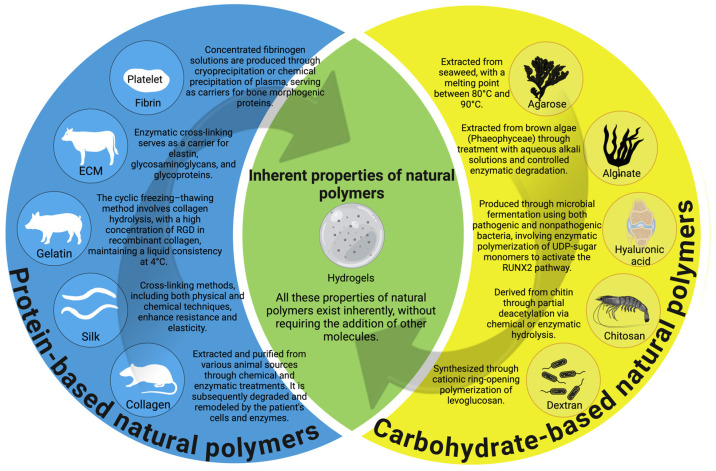
HGs derived from natural polymers, such as proteins or carbohydrates, exhibit distinct properties inherent to their molecular structure. These intrinsic characteristics confer these HGs (except alginate) osteoconductive capacities that support resident cell populations, eliminating the necessity for additional molecular modifications or supplements.

**Table 2 gels-11-00175-t002:** Natural polymer-based HGs.

Natural Polymer-Based HG	Properties	Applications	Cross-Linking	References
Collagen	Biocompatibility Biodegradability Cationic polymer Ready degradation High cross-linking potential Mechanical compression	Tissue engineering Bone regeneration	Enzymatic cross-linking	[3,16,17,21]
Fibrin	Biocompatibility Biodegradability Injectable application Elastic and plasticity Minimal inflammatory reaction	Drug carrier Bone regeneration Osteogenic capability Regeneration cartilage Skin tissues	Enzymatic cross-linking	[16,77,78]
Gelatin	Negative charge Chemical strength	Drug carrier Cartilage regeneration	Hydrophobic cross-linking	[17,36,79,80]
Extracellular bone matrix (ECM)	Antibacterial activity support of essential molecules Maintain homeostasis Easily mixed with other materials	Bone regeneration Organoid formation Dental implants survival Drug carrier	Enzymatic cross-linking	[81,82,83,84]
Agarose	Biocompatibility Biodegradability Antibacterial activity conductive activity Porous structure Water insoluble No need for cross-linked agents	Tissue engineering	Self-gelling at low temperatures	[17,21,85,86]
Alginate	Biocompatibility Biodegradability Water solubility High fluid absorption Regulated homeostasis Adhesive properties Stable 3D structure Rigidity	Tissue engineering	Water cross-linking	[87,88,89,90]
Hyaluronic acid	Antibacterial activity anionic copolymer Injectable Regulated cell interaction Anti-inflammatory Antiedematous Low toxicity Hypoallergenic	Tissue engineering Wound healing Drug carrier Bone regeneration	Multifunctional derivative cross-linking	[17,21,91,92,93]
Chitosan	Biocompatibility Biodegradability Antibacterial Antifungal Non-toxicity Adhesive properties polycationic polymer Hydrophilic Injectable	Hemostasis Wound healing Transdermal patches Skin, cartilage, and bone regeneration Blood vessel embolization Drug delivery Gene expression modulation	Ionic cross-linking in acid solution	[17,21,27,44,94,95]
Dextran	Biocompatibility Biodegradability injectable Non-toxic Easy chemical modification	Tissue regeneration Drug carrier	Free radical polymerization cross-linking	[96,97,98,99,100]
Silk	Hypoallergenic Resistance and elasticity Cell adhesion	Tissue regeneration Bone regeneration Wound healing Cartilage and tendon regeneration Drug carrier	Hydrophobic interactions cross-linked	[16,28]
Silk fibroin	Biocompatibility Controllable degradation Mechanical properties	Tissue engineering Cell and drug carriers	Self-assembly Shear force application Ultrasonication Electric field exposure cross-linking	[101,102,103]

**Table 4 gels-11-00175-t004:** Biomodels used for the evaluation of bone regeneration with HG.

Biomodel	HG	Bone Defect	In Vivo Evaluation	In Vivo Results	References
Rodents biomodels					
Sprague Dawley rats	Composite of chitosan microspheres loaded with BMP-2 embedded within a type I collagen HG	Critical cranial defects	µCT Histological analysis	Significant volume of newly formed bone eight weeks after	[110]
	Gelatin with black phosphorus nanosheet and deferoxamine	Defects in tibial bone	Macroscopic and histological analysis	Satisfactory results 4 weeks after administration of the material on 3 mm diameter defects in tibial bone	[118]
	Silk fibroin and fibrin with amorphous calcium phosphate and platelet-rich plasma HG	Critical cranial defects	µCT	New bone formation was observed 8 weeks	[127]
	Alginate loaded with allogeneic BMSCs HG	Cleft lip and palate injuries	µCT Histological sections at different time points	Regenerate bone defects	[87]
	PEG and PCL loaded with solid lipid nanoparticles HG	Critical cranial defects	µCT	Regenerate bone defects	[135]
	Chitosan microspheres containing basic fibroblast growth factor (bFGF) and transforming growth factor β3 (TGFβ3) HG	Alveolar bone defects Spherical defect with a diameter of 2 mm and depth of 1 mm	µCT	Successfully bone regeneration	[140]
	PEG with RGD cell adhesion motifs and MBP2 HG	Critical cranial defects	µCT Histological analysis	Bone regeneration after 4 weeks	[168]
	Collagen hydroxyapatite with BMP-2 and microspheres with alendronate	Critical cranial defects	µCT Histological analysis	Accumulation and maturation after 2, 4, and 8 weeks	[172]
	Amorphous calcium phosphate nanoparticle–platelet injectable HG	Lower and middle parts of the femur with 2.4 mm diameter	µCT Histological analysis	Significant improvements in tissue density after 2 weeks	[19]
	Gelatin/chitosan cryogel with platelet derived growth factor and BMP-2	Cranial bone defects	µCT	After 4 weeks, notable degree of new bone formation After 8 weeks, more than 50% of the initial bone defect was cover with new bone	[5]
Wistar rats	Fibrin HG combined with BMP-2 (37.5 µg/mL)	Tibial defect	µCT	Significant increases in total bone volume	[77]
	Chitosan derivative/collagen thermosensitive composite loads with fibroblastic growth factor	Infrabony periodontal defects	Hematoxylin–eosin staining Masson staining	Increase in alveolar bone height	[173]
	Gelatin with oxidized chondroitin sulfate and mesoporous bioactive glass nanoparticles	Cranial bone defects	µCT	Facilitates the regeneration	[116]
	Chitosan PEG with mineral-coated microparticle BMP-2	Cranial bone defects	µCT Histological analysis	Bone volume and tissue volume after 4 and 8 weeks	[174]
C57BL/6J mice	Dextran–tyramine conjugate with fibroblastic growth factor HG	Femur fracture	µCT	Satisfactory bone formation	[145]
	Photocross-linked dextran-based HGs loaded with human umbilical vein endothelial cells	Cranial bone defects	µCt Histology Immunohistology	Potential bone regeneration	[146]
BALB/c mice	Collagen with γ-PGA loaded with BMP-2 HG	Cranial bone defects	µCT Histological analysis	Potential bone regeneration	[167]
Fisher 344jcl rats	Resorbable collagen membrane with fibroblastic growth factor HG	4 mm mandibular bone defects	µCT Histological analysis	Volume of newly formed bone, bone mineral density and closure percentage of the defect after 6 weeks	[175]
Larger animals biomodels					
New Zealand rabbits	Extracellular matrix from porcine with biphasic calcium phosphate powder HG	Femoral head defect	µCT Immunohistochemistry Histological analysis	No immune responses, and new bone formation after 4 weeks After 8 weeks, no fibrous tissue bridge was evident	[123]
	Injectable rhBMP-2-loaded chitosan HG	Radius critical size defect	Radiographic evaluation Histological analysis	Discontinued bone was observed after 8 weeks	[139]
	Gelatin with BMP-9-coated nano-calcium-deficient hydroxyapatite/poly-amino acid	Femoral bone defects	X-ray observation µCT SEM	Satisfactory bone regeneration assessed at 1, 4, 12, and 24 weeks	[176]
Monkeys					
Cynomolgus monkeys	bFGF-incorporating gelatin HG	Four defects (each 6 mm in diameter) in each monkey’s parietal bone	Dual-energy X-ray absorptiometry (DEXA)	Defects are visible radiographically until week 18 Finally, at week 21, defects are observed with radiopacity	[177]

**Table 5 gels-11-00175-t005:** Key components and considerations for hydrogel-based bone regeneration: cell sources, biomaterials, bioactive factors, functional assessments, and potential challenges.

	**Human derived**		
**Mesenchymal stem cells used in bone regeneration**	Bone marrow-derived mesenchymal stem cells (BMMSCs)	Human mesenchymal stem cells (hMSCs)	Human adipose-derived stem cells (hASCs)
	Urine-derived stem cells (USCs)	Nasal mucosa-derived ectodermal mesenchymal stem cells (EMSCs)	Dental pulp stem cells (DPSCs)
	Human umbilical vein endothelial cells (HUVECs).	Human amnion-derived stem cells (HAMSCs)	
	**Animal derived**		
	Murine dental pulp stem cells (mDPSCs)	Rat-derived mesenchymal stem cells (rMSCs)	Bone marrow derived from rabbit mesenchymal stem cells (rBMMSCs)
	Murine dental pulp stem cells (mDPSCs)		
	**Human derived**		
**Cell lines used for in vitro evaluations in bone regeneration**	Endothelial cells	MC3T3-E1 pre-osteoblasts	Human osteoblast (hFOB)
	Macrophages isolated from peripheral blood	Human dermal fibroblasts (HDFa)	Human osteosarcoma cell cultures (SaOS-2)
	Human vascular endothelial cells (HUVECs)	Osteosarcoma (MG-63)	
	**Animal derived**		
	Bone stromal cell cultures (W20-17)	Murine macrophages (RAW 264.7)	KUSA-A1 cells
**Assays to evaluate the osteogenic capacities of Hg**	Alkaline phosphatase (ALP) activity	µCT density	Quantitative real-time PCR
	Calcium depositions with Alizarin Red staining	Critical bone defects	MTT
	Live/dead staining	M1 and M2 phenotypes	Activating signaling pathways associated with Yes-Associated Protein (YAP)
	Fluorescence microscopy	Cellular migration	
**Biomaterials to combine with HGs in bone regeneration**	Konjac glucomannan	Salts such as calcium and sodium	Platelet-rich plasma (PRP)
	Oxidized chondroitin sulfate	Mesoporous bioactive glass (BAG) nanoparticles	Biphasic calcium phosphate
	β-tricalcium phosphate	Laponite	Amorphous calcium phosphate (ACP
	Magnesium phosphate cement (MPC)	Calcium phosphate particles	Curcumin
	Zeolitic imidazolate	Flavonoid nanoparticles	Bone powder
	Poly(L-lysine)	Zn	Cu
	Mg	Solid lipid nanoparticles	Hydroxyapatite
**Growth factors and peptides used in bone regeneration**	Bone morphogenetic protein (BMP-2)	Bone morphogenetic protein (BMP-9)	Parathyroid hormone-related peptide (PTHrP
	Transforming growth factor β3 (TGFβ3)	Basic fibroblast growth factor (bFGF)	Stromal-derived growth factor (SDF)
	Platelet is a potent source of growth factors like (PDGF)	Osteogenic growth peptide (OGP)	Arginine–glycine–aspartic acid (RGD
	LL37	Peptide MP196	Fluorenylmethoxycarbonyl-diphenylalanine
**Possible negative effects of HGs on bone regeneration**	Ectopic bone in muscle tissue and joints	Epithelial tissue overgrowth	Fibrotic encapsulation
	High-production-cost peptides and growth factors		
**Genes related to osteogenic capacities**	Runxs 2,	Collagen type I (CoL-I)	BMP-4,
	Bone Sialoprotein (BSP)	Osteocalcin (OCN)	Osteopontin (OPN)
	Osterix (Osx)	STRO-1 gene	ALP

## Data Availability

No new data were created or analyzed in this study. Data sharing is not applicable to this article.

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
