# Peer review of "Hydrogel-Based Scaffolds: Advancing Bone Regeneration Through Tissue Engineering"

_gels, 2025, doi:10.3390/gels11030175_

Round 1

Reviewer 1 Report (New Reviewer)

Comments and Suggestions for Authors

The manuscript provides a well-organized review of hydrogels used in bone tissue engineering, particularly in their classification and role in the bone healing process. The inclusion of recent studies and the comparative analysis of hydrogel types would be valuable to readers exploring biomaterial-based strategies for bone regeneration.

However, there are aspects to improve. Specifically:

Alignment Between Sections:

Section 6 introduces concepts such as ectopic bone formationhydrogel stability, and immune response, which were not addressed in earlier sections. To strengthen coherence, we recommend either expanding the methodology/results to include data or analysis on these topics, or revising the discussion to focus exclusively on themes supported by the current content (e.g., key issues in the selection of hydrogels).

The conclusions emphasize the importance of mechanical and bioactive properties in hydrogels. However, these aspects were not analyzed in detail in the manuscript. I sugest to add a dedicated section evaluating these properties (e.g., elasticity, compressive strength, osteoinductive capabilities), or adjust the conclusions to reflect the manuscript’s actual focus (e.g., hydrogel design criteria).

Contextualization with Recent Literature:

While the review is thorough, it does not sufficiently distinguish itself from recent publications on the same topic, for example:

  • Xiong, Xiong (2022). Journal of Materials Science, 57, 887-913.

  • Ding et al. (2023). Molecules, 28(20), 7039.

 To enhance novelty, consider incorporating understudied angles (e.g., clinical translation challenges, cost-effectiveness, or scalability of hydrogels) or adding information of specific papers (e.g., comparative performance tables, meta-analysis of in vivo outcomes) if feasible.

I appreciate the effort invested in this work and believe these revisions will enhance its impact and relevance to the field.

Author Response

We sincerely appreciate the reviewers and the editor for their valuable work and, above all, for their insightful recommendations regarding this manuscript.

REVIEWER 1

the manuscript provides a well-organized review of hydrogels used in bone tissue engineering, particularly in their classification and role in the bone healing process. The inclusion of recent studies and the comparative analysis of hydrogel types would be valuable to readers exploring biomaterial-based strategies for bone regeneration.

However, there are aspects to improve. Specifically:

Alignment Between Sections:

Section 6 introduces concepts such as ectopic bone formationhydrogel stability, and immune response, which were not addressed in earlier sections. To strengthen coherence, we recommend either expanding the methodology/results to include data or analysis on these topics, or revising the discussion to focus exclusively on themes supported by the current content (e.g., key issues in the selection of hydrogels).

 Thank you for your valuable feedback. To address the need for greater coherence, we have incorporated a new section (Section 4) titled "Key Issues in the Selection of Hydrogels", as you suggested. This section outlines critical properties hydrogels (HGs) should possess to effectively support bone regeneration. For instance, it emphasizes the importance of integrating mesenchymal stem cells (MSCs) and bioactive peptides, which not only promote favorable immune responses but also help minimize inflammation and reduce the risk of material rejection. Additionally, the section highlights potential challenges associated with hydrogel applications, such as ectopic bone formation, inflammatory reactions, and fibrotic encapsulation of implanted materials. To address these issues, we discuss recent advancements and proposed strategies to mitigate these challenges, including surface modification techniques and controlled release of anti-inflammatory agents.To ensure the section is evidence-based, we have included five recent citations that support the discussed mechanisms, solutions, and innovations in hydrogel design for bone regeneration.

Zhang, Y., et al., Establishment of immortalized rabbit bone marrow mesenchymal stem cells and a preliminary study of their osteogenic differentiation capability. Animal Model Exp Med, 2024. 7(6): p. 824-834.

Yang, J., et al., Inflammation-Responsive Hydrogel Spray for Synergistic Prevention of Traumatic Heterotopic Ossification via Dual-Homeostatic Modulation Strategy. Adv Sci (Weinh), 2023. 10(30): p. e2302905.

Zhuo, H., et al., Gelatin methacryloyl @MP196/exos hydrogel induced neutrophil apoptosis and macrophage M2 polarization to inhibit periodontal bone loss. Colloids Surf B Biointerfaces, 2025. 248: p. 114466.

Zhou, X., et al., Poly(Glutamic Acid-Lysine) Hydrogels with Alternating Sequence Resist the Foreign Body Response in Rodents and Non-Human Primates. Adv Sci (Weinh), 2024. 11(16): p. e2308077.

Zhao, T., et al., Effect of injectable calcium alginate-amelogenin hydrogel on macrophage polarization and promotion of jawbone osteogenesis. RSC Adv, 2024. 14(3): p. 2016-2026.

The conclusions emphasize the importance of mechanical and bioactive properties in hydrogels. However, these aspects were not analyzed in detail in the manuscript. I sugest to add a dedicated section evaluating these properties (e.g., elasticity, compressive strength, osteoinductive capabilities), or adjust the conclusions to reflect the manuscript’s actual focus (e.g., hydrogel design criteria).

Thank you for your insightful comment. To address the need for more comprehensive data, we have introduced two new subsections: "Physical Properties of Hydrogels" (Section 5) and "Bioactive Properties of Hydrogels" (Section 6).

In Section 5, we examine the physical characteristics of hydrogels (HGs), beginning with their cross-linking mechanisms and how these determine key structural properties. The subsection also explores how hydrogel stiffness modulates the proliferative potential of mesenchymal stem cells (MSCs), a critical factor in bone regeneration. Furthermore, we highlight studies demonstrating how porosity enhances cellular activities such as proliferation and migration. Finally, we discuss water absorption capacity and the compositional requirements for achieving this property in HGs.

Section 6 focuses on three essential bioactive properties of hydrogels: Osteogenic proliferative capacity, which supports bone tissue formation, Controlled drug release capability, enabling localized delivery of therapeutic agents, Bioactive signaling, which promotes interactions with surrounding tissues.

To ensure the content is supported by the latest research, 15 recent references have been incorporated across these sections. These citations validate the discussed mechanisms, design strategies, and functional advantages of hydrogels in bone regeneration applications:

Zuev, Y.F., et al., Water as a Structural Marker in Gelatin Hydrogels with Different Cross-Linking Nature. Int J Mol Sci, 2024. 25(21).

Li, P., et al., Simple Preparation and Bone Regeneration Effects of Poly(vinyl alcohol)-Resveratrol Self-Cross-Linked Hydrogels. ACS Omega, 2024. 9(50): p. 49043-49053.

Zhu, M., et al., Hydrogel-based microenvironment engineering of haematopoietic stem cells. Cell Mol Life Sci, 2023. 80(2): p. 49.

Xu, D., et al., Depth profiling via nanoindentation for characterisation of the elastic modulus and hydraulic properties of thin hydrogel layers. J Mech Behav Biomed Mater, 2023. 148: p. 106195.

Wu, Y., et al., Multiscale design of stiffening and ROS scavenging hydrogels for the augmentation of mandibular bone regeneration. Bioact Mater, 2023. 20: p. 111-125.

Wang, Z., et al., Bioactive prosthesis interface compositing variable-stiffness hydrogels regulates stem cells fates to facilitate osseointegration through mechanotransduction. Int J Biol Macromol, 2024. 259(Pt 2): p. 129073.

Xue, J., et al., Hydrogel Composite Magnetic Scaffolds: Toward Cell-Free In Situ Bone Tissue Engineering. ACS Appl Bio Mater, 2024. 7(1): p. 168-181.

Arpornmaeklong, P., et al., Effects of Calcium Carbonate Microcapsules and Nanohydroxyapatite on Properties of Thermosensitive Chitosan/Collagen Hydrogels. Polymers (Basel), 2023. 15(2).

Abudukelimu, K., et al., Preliminary study on the preparation of antler powder/chitosan/β-glycerophosphate sodium/polyvinyl alcohol porous hydrogel scaffolds and their osteogenic effects. Front Bioeng Biotechnol, 2024. 12: p. 1421718.

Zhu, Y., et al., Multifunctional injectable hydrogel system as a mild photothermal-assisted therapeutic platform for programmed regulation of inflammation and osteo-microenvironment for enhanced healing of diabetic bone defects in situ. Theranostics, 2024. 14(18): p. 7140-7198.

Wang, X., et al., Recent Strategies and Advances in Hydrogel-Based Delivery Platforms for Bone Regeneration. Nanomicro Lett, 2024. 17(1): p. 73.

Zhu, W., et al., Fabrication of Naturally Derived Double-Network Hydrogels With a Sustained Aspirin Release System for Facilitating Bone Regeneration. Front Chem, 2022. 10: p. 874985.

Zhu, L., et al., Flexible Fabrication and Hybridization of Bioactive Hydrogels with Robust Osteogenic Potency. Pharmaceutics, 2023. 15(10).

Zheng, L., et al., Oscillating Fluid Flow Activated Osteocyte Lysate-Based Hydrogel for Regulating Osteoblast/Osteoclast Homeostasis to Enhance Bone Repair. Adv Sci (Weinh), 2023. 10(15): p. e2204592.

Zhang, W., et al., Construction of artificial periosteum with methacrylamide gelatin hydrogel-wharton's jelly based on stem cell recruitment and its application in bone tissue engineering. Mater Today Bio, 2023. 18: p. 100528.

Soleymani, H., et al., Single-layer graphene oxide nanosheets induce proliferation and Osteogenesis of single-cell hBMSCs encapsulated in Alginate Microgels. Sci Rep, 2024. 14(1): p. 25272.

Contextualization with Recent Literature:

While the review is thorough, it does not sufficiently distinguish itself from recent publications on the same topic, for example:

  • Xiong, Xiong (2022). Journal of Materials Science, 57, 887-913.
  • Ding et al. (2023). Molecules, 28(20), 7039.

Thank you very much for the comment.

In order to try to differentiate our work with those mentioned below

Xiong, Xiong (2022). Journal of Materials Science, 57, 887-913.

Ding et al. (2023). Molecules, 28(20), 7039.

We have added 4 new tables trying to express relevant information such as the main biomodels used in the field of in vivo bone regeneration.

In addition, in total, with all our observations and reviews, we have gone from a total of 171 references to a total of 205, which means a total of 34 new citations with a period of no more than 3 years.

To enhance novelty, consider incorporating understudied angles (e.g., clinical translation challenges, cost-effectiveness, or scalability of hydrogels) or adding information of specific papers (e.g., comparative performance tables, meta-analysis of in vivo outcomes) if feasible.

Thank you very much for the advice, to expose and be able to express to the reader different topics and perspectives such as the physical and chemical responses of Hg we have added table number 1 where we explain the responses to temperature, photo light and electro and magnetic responses in addition to pH, glucose and biological and chemical responses. Data extracted from the following quote

Mantha, S., et al., Smart Hydrogels in Tissue Engineering and Regenerative Medicine. Materials (Basel), 2019. 12(20).

We also added table number 2 where we express the main MSCs of human and animal origin that are used in the field of bone regeneration through HG, the main cell lines for in vitro tests as well as the tests that are used to evaluate osteogenic, mineralizing, cytotoxicity, inflammation and immunological responses capacities among other important aspects.

Finally, we have added a section (Biomodels, type of bone defect and tests for the evaluation of bone defect regeneration) and a new table where we describe the reported in vivo tests, categorizing the biomodels in rodents and large animals such as rabbits and monkeys, in addition to mentioning the type of bone defect and tests for its evaluation.

New references added

Zhou, B., et al., GelMA-based bioactive hydrogel scaffolds with multiple bone defect repair functions: therapeutic strategies and recent advances. Biomater Res, 2023. 27(1): p. 86.

Qi, J., H. Wu, and G. Liu, Novel Strategies for Spatiotemporal and Controlled BMP-2 Delivery in Bone Tissue Engineering. Cell Transplant, 2024. 33: p. 9636897241276733.

Mantha, S., et al., Smart Hydrogels in Tissue Engineering and Regenerative Medicine. Materials (Basel), 2019. 12(20).

Zhang, Y., et al., Establishment of immortalized rabbit bone marrow mesenchymal stem cells and a preliminary study of their osteogenic differentiation capability. Animal Model Exp Med, 2024. 7(6): p. 824-834.

Yang, J., et al., Inflammation-Responsive Hydrogel Spray for Synergistic Prevention of Traumatic Heterotopic Ossification via Dual-Homeostatic Modulation Strategy. Adv Sci (Weinh), 2023. 10(30): p. e2302905.

Zhuo, H., et al., Gelatin methacryloyl @MP196/exos hydrogel induced neutrophil apoptosis and macrophage M2 polarization to inhibit periodontal bone loss. Colloids Surf B Biointerfaces, 2025. 248: p. 114466.

Zhao, T., et al., Effect of injectable calcium alginate-amelogenin hydrogel on macrophage polarization and promotion of jawbone osteogenesis. RSC Adv, 2024. 14(3): p. 2016-2026.

Zhou, X., et al., Poly(Glutamic Acid-Lysine) Hydrogels with Alternating Sequence Resist the Foreign Body Response in Rodents and Non-Human Primates. Adv Sci (Weinh), 2024. 11(16): p. e2308077.

Zuev, Y.F., et al., Water as a Structural Marker in Gelatin Hydrogels with Different Cross-Linking Nature. Int J Mol Sci, 2024. 25(21).

Li, P., et al., Simple Preparation and Bone Regeneration Effects of Poly(vinyl alcohol)-Resveratrol Self-Cross-Linked Hydrogels. ACS Omega, 2024. 9(50): p. 49043-49053.

Zhu, M., et al., Hydrogel-based microenvironment engineering of haematopoietic stem cells. Cell Mol Life Sci, 2023. 80(2): p. 49.

Xu, D., et al., Depth profiling via nanoindentation for characterisation of the elastic modulus and hydraulic properties of thin hydrogel layers. J Mech Behav Biomed Mater, 2023. 148: p. 106195.

Wu, Y., et al., Multiscale design of stiffening and ROS scavenging hydrogels for the augmentation of mandibular bone regeneration. Bioact Mater, 2023. 20: p. 111-125.

Wang, Z., et al., Bioactive prosthesis interface compositing variable-stiffness hydrogels regulates stem cells fates to facilitate osseointegration through mechanotransduction. Int J Biol Macromol, 2024. 259(Pt 2): p. 129073.

Xue, J., et al., Hydrogel Composite Magnetic Scaffolds: Toward Cell-Free In Situ Bone Tissue Engineering. ACS Appl Bio Mater, 2024. 7(1): p. 168-181.

Arpornmaeklong, P., et al., Effects of Calcium Carbonate Microcapsules and Nanohydroxyapatite on Properties of Thermosensitive Chitosan/Collagen Hydrogels. Polymers (Basel), 2023. 15(2).

Abudukelimu, K., et al., Preliminary study on the preparation of antler powder/chitosan/β-glycerophosphate sodium/polyvinyl alcohol porous hydrogel scaffolds and their osteogenic effects. Front Bioeng Biotechnol, 2024. 12: p. 1421718.

Zhu, Y., et al., Multifunctional injectable hydrogel system as a mild photothermal-assisted therapeutic platform for programmed regulation of inflammation and osteo-microenvironment for enhanced healing of diabetic bone defects in situ. Theranostics, 2024. 14(18): p. 7140-7198.

Wang, X., et al., Recent Strategies and Advances in Hydrogel-Based Delivery Platforms for Bone Regeneration. Nanomicro Lett, 2024. 17(1): p. 73.

Zhu, W., et al., Fabrication of Naturally Derived Double-Network Hydrogels With a Sustained Aspirin Release System for Facilitating Bone Regeneration. Front Chem, 2022. 10: p. 874985.

Zhu, L., et al., Flexible Fabrication and Hybridization of Bioactive Hydrogels with Robust Osteogenic Potency. Pharmaceutics, 2023. 15(10).

Zheng, L., et al., Oscillating Fluid Flow Activated Osteocyte Lysate-Based Hydrogel for Regulating Osteoblast/Osteoclast Homeostasis to Enhance Bone Repair. Adv Sci (Weinh), 2023. 10(15): p. e2204592.

Zhang, W., et al., Construction of artificial periosteum with methacrylamide gelatin hydrogel-wharton's jelly based on stem cell recruitment and its application in bone tissue engineering. Mater Today Bio, 2023. 18: p. 100528.

Yu, C., et al., GelMA hydrogels reinforced by PCL@GelMA nanofibers and bioactive glass induce bone regeneration in critical size cranial defects. J Nanobiotechnology, 2024. 22(1): p. 696.

Zhang, X., et al., A Bioactive Gelatin-Methacrylate Incorporating Magnesium Phosphate Cement for Bone Regeneration. Biomedicines, 2024. 12(1).

Yuan, S., et al., Immunoregulation in Skull Defect Repair with a Smart Hydrogel Loaded with Mesoporous Bioactive Glasses. Biomater Res, 2024. 28: p. 0074.

Yu, T., et al., An injectable and self-healing hydrogel with dual physical crosslinking for in-situ bone formation. Mater Today Bio, 2023. 19: p. 100558.

Neumann, M., et al., Stimuli-Responsive Hydrogels: The Dynamic Smart Biomaterials of Tomorrow. Macromolecules, 2023. 56(21): p. 8377-8392.

Reviewer 2 Report (New Reviewer)

Comments and Suggestions for Authors

The authors have compiled a review manuscript that includes many of the most recent manuscripts based on the content of the title. The reviewer carefully read the submitted manuscript and found the content described to be well organized with respect to bone regeneration, HG scaffolds involved in tissue regeneration, and descriptions of various natural and synthetic polymers. The reviewer found it to be a very excellent review manuscript.

Therefore, the reviewer suggests that the following two points be added to the review manuscript to make it a review manuscript that attracts more attention from readers.

  1. On page 5, line 184, the authors describe mesenchymal stem cells (MSCs). Therefore, please briefly introduce that bone regenerative medicine using MSCs is already in clinical practice. Please also mention that MSCs secrete various growth factors described in the second half of the manuscript, and that MSCs not only differentiate into bone cells on their own (autocrine), but also perform tissue regeneration and repair (paracrine) with various growth factors secreted by MSCs.
  2. For HG scaffolds, please briefly introduce that substances such as those mentioned in the section on growth factors can be expected to have a sustained release effect when properly formulated with polymers, resulting in improved therapeutic efficacy.

Author Response

We sincerely appreciate the reviewers and the editor for their valuable work and, above all, for their insightful recommendations regarding this manuscript.

The authors have compiled a review manuscript that includes many of the most recent manuscripts based on the content of the title. The reviewer carefully read the submitted manuscript and found the content described to be well organized with respect to bone regeneration, HG scaffolds involved in tissue regeneration, and descriptions of various natural and synthetic polymers. The reviewer found it to be a very excellent review manuscript.

Therefore, the reviewer suggests that the following two points be added to the review manuscript to make it a review manuscript that attracts more attention from readers.

On page 5, line 184, the authors describe mesenchymal stem cells (MSCs). Therefore, please briefly introduce that bone regenerative medicine using MSCs is already in clinical practice. Please also mention that MSCs secrete various growth factors described in the second half of the manuscript, and that MSCs not only differentiate into bone cells on their own (autocrine), but also perform tissue regeneration and repair (paracrine) with various growth factors secreted by MSCs.

Thank you very much for your careful observation. The description of Figure 1 is rewritten according to your observations, emphasizing that MSCs have been studied and used in clinical trials in addition to the immunological basis (autocrine and paracrine).

For HG scaffolds, please briefly introduce that substances such as those mentioned in the section on growth factors can be expected to have a sustained release effect when properly formulated with polymers, resulting in improved therapeutic efficacy.

Many thanks for the comment. We have added in section 3 “Hydrogels-based scaffolds and their role in tissue engineering” a brief description of the importance of the use of the growth factor BMP emphasizing that the nature of its incorporation by covalent bonds or not substantially influences the behavior of the release of the growth factor.

The following references are added

Zhou, B., et al., GelMA-based bioactive hydrogel scaffolds with multiple bone defect repair functions: therapeutic strategies and recent advances. Biomater Res, 2023. 27(1): p. 86.

Qi, J., H. Wu, and G. Liu, Novel Strategies for Spatiotemporal and Controlled BMP-2 Delivery in Bone Tissue Engineering. Cell Transplant, 2024. 33: p. 9636897241276733.

New reference added in general to improve the mauscript

Zhou, B., et al., GelMA-based bioactive hydrogel scaffolds with multiple bone defect repair functions: therapeutic strategies and recent advances. Biomater Res, 2023. 27(1): p. 86.

Qi, J., H. Wu, and G. Liu, Novel Strategies for Spatiotemporal and Controlled BMP-2 Delivery in Bone Tissue Engineering. Cell Transplant, 2024. 33: p. 9636897241276733.

Mantha, S., et al., Smart Hydrogels in Tissue Engineering and Regenerative Medicine. Materials (Basel), 2019. 12(20).

Zhang, Y., et al., Establishment of immortalized rabbit bone marrow mesenchymal stem cells and a preliminary study of their osteogenic differentiation capability. Animal Model Exp Med, 2024. 7(6): p. 824-834.

Yang, J., et al., Inflammation-Responsive Hydrogel Spray for Synergistic Prevention of Traumatic Heterotopic Ossification via Dual-Homeostatic Modulation Strategy. Adv Sci (Weinh), 2023. 10(30): p. e2302905.

Zhuo, H., et al., Gelatin methacryloyl @MP196/exos hydrogel induced neutrophil apoptosis and macrophage M2 polarization to inhibit periodontal bone loss. Colloids Surf B Biointerfaces, 2025. 248: p. 114466.

Zhao, T., et al., Effect of injectable calcium alginate-amelogenin hydrogel on macrophage polarization and promotion of jawbone osteogenesis. RSC Adv, 2024. 14(3): p. 2016-2026.

Zhou, X., et al., Poly(Glutamic Acid-Lysine) Hydrogels with Alternating Sequence Resist the Foreign Body Response in Rodents and Non-Human Primates. Adv Sci (Weinh), 2024. 11(16): p. e2308077.

Zuev, Y.F., et al., Water as a Structural Marker in Gelatin Hydrogels with Different Cross-Linking Nature. Int J Mol Sci, 2024. 25(21).

Li, P., et al., Simple Preparation and Bone Regeneration Effects of Poly(vinyl alcohol)-Resveratrol Self-Cross-Linked Hydrogels. ACS Omega, 2024. 9(50): p. 49043-49053.

Zhu, M., et al., Hydrogel-based microenvironment engineering of haematopoietic stem cells. Cell Mol Life Sci, 2023. 80(2): p. 49.

Xu, D., et al., Depth profiling via nanoindentation for characterisation of the elastic modulus and hydraulic properties of thin hydrogel layers. J Mech Behav Biomed Mater, 2023. 148: p. 106195.

Wu, Y., et al., Multiscale design of stiffening and ROS scavenging hydrogels for the augmentation of mandibular bone regeneration. Bioact Mater, 2023. 20: p. 111-125.

Wang, Z., et al., Bioactive prosthesis interface compositing variable-stiffness hydrogels regulates stem cells fates to facilitate osseointegration through mechanotransduction. Int J Biol Macromol, 2024. 259(Pt 2): p. 129073.

Xue, J., et al., Hydrogel Composite Magnetic Scaffolds: Toward Cell-Free In Situ Bone Tissue Engineering. ACS Appl Bio Mater, 2024. 7(1): p. 168-181.

Arpornmaeklong, P., et al., Effects of Calcium Carbonate Microcapsules and Nanohydroxyapatite on Properties of Thermosensitive Chitosan/Collagen Hydrogels. Polymers (Basel), 2023. 15(2).

Abudukelimu, K., et al., Preliminary study on the preparation of antler powder/chitosan/β-glycerophosphate sodium/polyvinyl alcohol porous hydrogel scaffolds and their osteogenic effects. Front Bioeng Biotechnol, 2024. 12: p. 1421718.

Zhu, Y., et al., Multifunctional injectable hydrogel system as a mild photothermal-assisted therapeutic platform for programmed regulation of inflammation and osteo-microenvironment for enhanced healing of diabetic bone defects in situ. Theranostics, 2024. 14(18): p. 7140-7198.

Wang, X., et al., Recent Strategies and Advances in Hydrogel-Based Delivery Platforms for Bone Regeneration. Nanomicro Lett, 2024. 17(1): p. 73.

Zhu, W., et al., Fabrication of Naturally Derived Double-Network Hydrogels With a Sustained Aspirin Release System for Facilitating Bone Regeneration. Front Chem, 2022. 10: p. 874985.

Zhu, L., et al., Flexible Fabrication and Hybridization of Bioactive Hydrogels with Robust Osteogenic Potency. Pharmaceutics, 2023. 15(10).

Zheng, L., et al., Oscillating Fluid Flow Activated Osteocyte Lysate-Based Hydrogel for Regulating Osteoblast/Osteoclast Homeostasis to Enhance Bone Repair. Adv Sci (Weinh), 2023. 10(15): p. e2204592.

Zhang, W., et al., Construction of artificial periosteum with methacrylamide gelatin hydrogel-wharton's jelly based on stem cell recruitment and its application in bone tissue engineering. Mater Today Bio, 2023. 18: p. 100528.

Yu, C., et al., GelMA hydrogels reinforced by PCL@GelMA nanofibers and bioactive glass induce bone regeneration in critical size cranial defects. J Nanobiotechnology, 2024. 22(1): p. 696.

Zhang, X., et al., A Bioactive Gelatin-Methacrylate Incorporating Magnesium Phosphate Cement for Bone Regeneration. Biomedicines, 2024. 12(1).

Yuan, S., et al., Immunoregulation in Skull Defect Repair with a Smart Hydrogel Loaded with Mesoporous Bioactive Glasses. Biomater Res, 2024. 28: p. 0074.

Yu, T., et al., An injectable and self-healing hydrogel with dual physical crosslinking for in-situ bone formation. Mater Today Bio, 2023. 19: p. 100558.

Neumann, M., et al., Stimuli-Responsive Hydrogels: The Dynamic Smart Biomaterials of Tomorrow. Macromolecules, 2023. 56(21): p. 8377-8392.

Reviewer 3 Report (New Reviewer)

Comments and Suggestions for Authors

This review paper provides a comprehensive review of hydrogel (HG)-based scaffolds for bone tissue engineering. It covers a wide range of topics, including the properties of hydrogels, their applications in bone regeneration, and the integration of bioactive molecules such as growth factors and peptides.

The advantage of this manuscript is that it covers both natural and synthetic hydrogels, including their properties, advantages, and limitations and contains new information on this topic. Moreover, it also emphasizes the functionalization of hydrogels with bioactive molecules to enhance osteoconductivity, osteoinductivity, and osteogenesis, which is a growing area of interest in the field. The manuscript discusses advanced strategies such as bioprinting, bio-inks, and the use of exosomes, which are emerging technologies in tissue engineering. These topics are not always covered in such depth in similar reviews.

It also highlights the role of hydrogels in mimicking the extracellular matrix (ECM) and their ability to support cell adhesion, proliferation, and differentiation, which is critical for bone regeneration.

The paper itself is comprehensive and well-structured. It provides a valuable resource for researchers and clinicians interested in this field.

Below is a detailed review of the manuscript for improvement:

-While the manuscript provides a thorough overview of hydrogel-based scaffolds, it lacks critical analysis of the limitations and challenges associated with these materials. For example, it does not sufficiently address the variability in degradation rates, potential immune responses, or the difficulty in scaling up production for clinical use.

-The quality of paper would improve if it contains discussion of the drawbacks of hydrogels, such as their mechanical weakness compared to native bone tissue.

- Although the manuscript mentions several in vivo studies, it does not provide a systematic analysis of their outcomes. A more detailed discussion of the efficacy, safety, and limitations of hydrogel-based scaffolds in animal models would strengthen the review.

-The manuscript could also include a comparison of different animal models (e.g., rodents vs. larger animals) and their relevance to human clinical applications.

-While natural polymers are discussed in detail, synthetic polymers get less attention. Given their tunable properties and potential for mass production, synthetic hydrogels need a more in-depth discussion, including their advantages and limitations compared to natural hydrogels.

- The "Future Perspective" section is brief and does not fully explore emerging trends and technologies in the field. For example, the manuscript could discuss the potential of smart hydrogels (e.g., stimuli-responsive hydrogels) or the integration of nanotechnology for enhanced drug delivery and mechanical properties.

- Some sections of the manuscript are repetitive, particularly in the discussion of growth factors and peptides. The organization could be improved to avoid redundancy and ensure a more streamlined presentation of information.

- The Authors should also include a section on the regulatory challenges and manufacturing considerations for hydrogel-based scaffolds, particularly in the context of scaling up production for clinical use.

- Addition of an overview of ongoing or completed clinical trials involving hydrogel-based scaffolds for bone regeneration, including their outcomes and lessons learned, would also improve the quality.

Comments on the Quality of English Language The English language needs some refinements.

Author Response

We sincerely appreciate the reviewers and the editor for their valuable work and, above all, for their insightful recommendations regarding this manuscript.

This review paper provides a comprehensive review of hydrogel (HG)-based scaffolds for bone tissue engineering. It covers a wide range of topics, including the properties of hydrogels, their applications in bone regeneration, and the integration of bioactive molecules such as growth factors and peptides.

The advantage of this manuscript is that it covers both natural and synthetic hydrogels, including their properties, advantages, and limitations and contains new information on this topic. Moreover, it also emphasizes the functionalization of hydrogels with bioactive molecules to enhance osteoconductivity, osteoinductivity, and osteogenesis, which is a growing area of interest in the field. The manuscript discusses advanced strategies such as bioprinting, bio-inks, and the use of exosomes, which are emerging technologies in tissue engineering. These topics are not always covered in such depth in similar reviews.

It also highlights the role of hydrogels in mimicking the extracellular matrix (ECM) and their ability to support cell adhesion, proliferation, and differentiation, which is critical for bone regeneration.

The paper itself is comprehensive and well-structured. It provides a valuable resource for researchers and clinicians interested in this field.

Below is a detailed review of the manuscript for improvement:

-While the manuscript provides a thorough overview of hydrogel-based scaffolds, it lacks critical analysis of the limitations and challenges associated with these materials. For example, it does not sufficiently address the variability in degradation rates, potential immune responses, or the difficulty in scaling up production for clinical use.

 We appreciate your observation. A new section (4) was added with the name proposed “key issues in the selection of hydrogels” where the main properties that hydrogels should have for bone regeneration are addressed, such as the use of MSCs and peptides that collaborate in immunological responses such as inflammation and rejection of the materials using 4 recent citations

Zhang, Y., et al., Establishment of immortalized rabbit bone marrow mesenchymal stem cells and a preliminary study of their osteogenic differentiation capability. Animal Model Exp Med, 2024. 7(6): p. 824-834.

Yang, J., et al., Inflammation-Responsive Hydrogel Spray for Synergistic Prevention of Traumatic Heterotopic Ossification via Dual-Homeostatic Modulation Strategy. Adv Sci (Weinh), 2023. 10(30): p. e2302905.

Zhuo, H., et al., Gelatin methacryloyl @MP196/exos hydrogel induced neutrophil apoptosis and macrophage M2 polarization to inhibit periodontal bone loss. Colloids Surf B Biointerfaces, 2025. 248: p. 114466.

Zhou, X., et al., Poly(Glutamic Acid-Lysine) Hydrogels with Alternating Sequence Resist the Foreign Body Response in Rodents and Non-Human Primates. Adv Sci (Weinh), 2024. 11(16): p. e2308077.

Zhao, T., et al., Effect of injectable calcium alginate-amelogenin hydrogel on macrophage polarization and promotion of jawbone osteogenesis. RSC Adv, 2024. 14(3): p. 2016-2026.

Also in section 10, data related to degradation problems were added, reporting the results of some in vitro tests

Yu, C., et al., GelMA hydrogels reinforced by PCL@GelMA nanofibers and bioactive glass induce bone regeneration in critical size cranial defects. J Nanobiotechnology, 2024. 22(1): p. 696.

Yuan, S., et al., Immunoregulation in Skull Defect Repair with a Smart Hydrogel Loaded with Mesoporous Bioactive Glasses. Biomater Res, 2024. 28: p. 0074.

-The quality of paper would improve if it contains discussion of the drawbacks of hydrogels, such as their mechanical weakness compared to native bone tissue.

Thank you very much for the observation. As mentioned in the previous observation, data on mechanical problems and difficulties are added in section 10. Data related to the inability to use injectable hydrogels for bone regeneration in load-bearing areas are discussed.

Zhang, X., et al., A Bioactive Gelatin-Methacrylate Incorporating Magnesium Phosphate Cement for Bone Regeneration. Biomedicines, 2024. 12(1).

Yu, T., et al., An injectable and self-healing hydrogel with dual physical crosslinking for in-situ bone formation. Mater Today Bio, 2023. 19: p. 100558

- Although the manuscript mentions several in vivo studies, it does not provide a systematic analysis of their outcomes. A more detailed discussion of the efficacy, safety, and limitations of hydrogel-based scaffolds in animal models would strengthen the review.

Thank you for your insightful comment. To address the need for more comprehensive data, we have introduced two new subsections: "Physical Properties of Hydrogels" (Section 5) and "Bioactive Properties of Hydrogels" (Section 6).

In Section 5, we examine the physical characteristics of hydrogels (HGs), beginning with their cross-linking mechanisms and how these determine key structural properties. The subsection also explores how hydrogel stiffness modulates the proliferative potential of mesenchymal stem cells (MSCs), a critical factor in bone regeneration. Furthermore, we highlight studies demonstrating how porosity enhances cellular activities such as proliferation and migration. Finally, we discuss water absorption capacity and the compositional requirements for achieving this property in HGs.

Section 6 focuses on three essential bioactive properties of hydrogels: Osteogenic proliferative capacity, which supports bone tissue formation, Controlled drug release capability, enabling localized delivery of therapeutic agents, Bioactive signaling, which promotes interactions with surrounding tissues.

To ensure the content is supported by the latest research, 15 recent references have been incorporated across these sections. These citations validate the discussed mechanisms, design strategies, and functional advantages of hydrogels in bone regeneration applications:

Zuev, Y.F., et al., Water as a Structural Marker in Gelatin Hydrogels with Different Cross-Linking Nature. Int J Mol Sci, 2024. 25(21).

Li, P., et al., Simple Preparation and Bone Regeneration Effects of Poly(vinyl alcohol)-Resveratrol Self-Cross-Linked Hydrogels. ACS Omega, 2024. 9(50): p. 49043-49053.

Zhu, M., et al., Hydrogel-based microenvironment engineering of haematopoietic stem cells. Cell Mol Life Sci, 2023. 80(2): p. 49.

Xu, D., et al., Depth profiling via nanoindentation for characterisation of the elastic modulus and hydraulic properties of thin hydrogel layers. J Mech Behav Biomed Mater, 2023. 148: p. 106195.

Wu, Y., et al., Multiscale design of stiffening and ROS scavenging hydrogels for the augmentation of mandibular bone regeneration. Bioact Mater, 2023. 20: p. 111-125.

Wang, Z., et al., Bioactive prosthesis interface compositing variable-stiffness hydrogels regulates stem cells fates to facilitate osseointegration through mechanotransduction. Int J Biol Macromol, 2024. 259(Pt 2): p. 129073.

Xue, J., et al., Hydrogel Composite Magnetic Scaffolds: Toward Cell-Free In Situ Bone Tissue Engineering. ACS Appl Bio Mater, 2024. 7(1): p. 168-181.

Arpornmaeklong, P., et al., Effects of Calcium Carbonate Microcapsules and Nanohydroxyapatite on Properties of Thermosensitive Chitosan/Collagen Hydrogels. Polymers (Basel), 2023. 15(2).

Abudukelimu, K., et al., Preliminary study on the preparation of antler powder/chitosan/β-glycerophosphate sodium/polyvinyl alcohol porous hydrogel scaffolds and their osteogenic effects. Front Bioeng Biotechnol, 2024. 12: p. 1421718.

Zhu, Y., et al., Multifunctional injectable hydrogel system as a mild photothermal-assisted therapeutic platform for programmed regulation of inflammation and osteo-microenvironment for enhanced healing of diabetic bone defects in situ. Theranostics, 2024. 14(18): p. 7140-7198.

Wang, X., et al., Recent Strategies and Advances in Hydrogel-Based Delivery Platforms for Bone Regeneration. Nanomicro Lett, 2024. 17(1): p. 73.

Zhu, W., et al., Fabrication of Naturally Derived Double-Network Hydrogels With a Sustained Aspirin Release System for Facilitating Bone Regeneration. Front Chem, 2022. 10: p. 874985.

Zhu, L., et al., Flexible Fabrication and Hybridization of Bioactive Hydrogels with Robust Osteogenic Potency. Pharmaceutics, 2023. 15(10).

Zheng, L., et al., Oscillating Fluid Flow Activated Osteocyte Lysate-Based Hydrogel for Regulating Osteoblast/Osteoclast Homeostasis to Enhance Bone Repair. Adv Sci (Weinh), 2023. 10(15): p. e2204592.

Zhang, W., et al., Construction of artificial periosteum with methacrylamide gelatin hydrogel-wharton's jelly based on stem cell recruitment and its application in bone tissue engineering. Mater Today Bio, 2023. 18: p. 100528.

Soleymani, H., et al., Single-layer graphene oxide nanosheets induce proliferation and Osteogenesis of single-cell hBMSCs encapsulated in Alginate Microgels. Sci Rep, 2024. 14(1): p. 25272.

-The manuscript could also include a comparison of different animal models (e.g., rodents vs. larger animals) and their relevance to human clinical applications.

Thank you very much for the recommendation. We have added a section (Biomodels, type of bone defect and tests for the evaluation of bone defect regeneration) and a new table where we describe the reported in vivo tests, categorizing the biomodels in rodents and large animals such as rabbits and monkeys, in addition to mentioning the type of bone defect and tests for its evaluation

-While natural polymers are discussed in detail, synthetic polymers get less attention. Given their tunable properties and potential for mass production, synthetic hydrogels need a more in-depth discussion, including their advantages and limitations compared to natural hydrogels.

We greatly appreciate your recommendation. We have added two synthetic polymers to strengthen this section in Table 4, in addition to their respective support in sections 7.2.4 7.2.4. Polyvinyl alcohol (PVA) and 7.2.5. Poly(γ-glutamic acid) (γ-PGA)

the added citations are the following

Li, P., et al., Simple Preparation and Bone Regeneration Effects of Poly(vinyl alcohol)-Resveratrol Self-Cross-Linked Hydrogels. ACS Omega, 2024. 9(50): p. 49043-49053.

Zheng, J., et al., Directed self-assembly of herbal small molecules into sustained release hydrogels for treating neural inflammation. Nat Commun, 2019. 10(1): p. 1604.

Huang, C.L., et al., Development of a flexible film made of polyvinyl alcohol with chitosan based thermosensitive hydrogel. J Dent Sci, 2023. 18(2): p. 822-832.

Xue, C., et al., Stimuli-responsive hydrogels for bone tissue engineering. Biomater Transl, 2024. 5(3): p. 257-273.

Baines, D.K., et al., The Enrichment of Whey Protein Isolate Hydrogels with Poly-γ-Glutamic Acid Promotes the Proliferation and Osteogenic Differentiation of Preosteoblasts. Gels, 2023. 10(1).

Cho, S.H., et al., In Situ-Forming Collagen/poly-γ-glutamic Acid Hydrogel System with Mesenchymal Stem Cells and Bone Morphogenetic Protein-2 for Bone Tissue Regeneration in a Mouse Calvarial Bone Defect Model. Tissue Eng Regen Med, 2022. 19(5): p. 1099-1111.

- The "Future Perspective" section is brief and does not fully explore emerging trends and technologies in the field. For example, the manuscript could discuss the potential of smart hydrogels (e.g., stimuli-responsive hydrogels) or the integration of nanotechnology for enhanced drug delivery and mechanical properties.

We really appreciate your recommendation. We have added the concept of smart hydrogel in the future perspective section with a brief explanation as well as the concept of 4d bioprinted

Mantha, S., et al., Smart Hydrogels in Tissue Engineering and Regenerative Medicine. Materials (Basel), 2019. 12(20).

Neumann, M., et al., Stimuli-Responsive Hydrogels: The Dynamic Smart Biomaterials of Tomorrow. Macromolecules, 2023. 56(21): p. 8377-8392.

- Addition of an overview of ongoing or completed clinical trials involving hydrogel-based scaffolds for bone regeneration, including their outcomes and lessons learned, would also improve the quality.

Thank you for the suggestion. We have added 4 new tables (1, 2, 4, 5) where different data related to hydrogels, characteristics, animal models for in vivo testing, growth factors and most commonly used peptides, among other relevant topics are expressed.In addition, in total, with all our observations and reviews, we have gone from a total of 171 references to a total of 205, which means a total of 34 new citations with a period of no more than 3 years.

New references added

Zhou, B., et al., GelMA-based bioactive hydrogel scaffolds with multiple bone defect repair functions: therapeutic strategies and recent advances. Biomater Res, 2023. 27(1): p. 86.

Qi, J., H. Wu, and G. Liu, Novel Strategies for Spatiotemporal and Controlled BMP-2 Delivery in Bone Tissue Engineering. Cell Transplant, 2024. 33: p. 9636897241276733.

Mantha, S., et al., Smart Hydrogels in Tissue Engineering and Regenerative Medicine. Materials (Basel), 2019. 12(20).

Zhang, Y., et al., Establishment of immortalized rabbit bone marrow mesenchymal stem cells and a preliminary study of their osteogenic differentiation capability. Animal Model Exp Med, 2024. 7(6): p. 824-834.

Yang, J., et al., Inflammation-Responsive Hydrogel Spray for Synergistic Prevention of Traumatic Heterotopic Ossification via Dual-Homeostatic Modulation Strategy. Adv Sci (Weinh), 2023. 10(30): p. e2302905.

Zhuo, H., et al., Gelatin methacryloyl @MP196/exos hydrogel induced neutrophil apoptosis and macrophage M2 polarization to inhibit periodontal bone loss. Colloids Surf B Biointerfaces, 2025. 248: p. 114466.

Zhao, T., et al., Effect of injectable calcium alginate-amelogenin hydrogel on macrophage polarization and promotion of jawbone osteogenesis. RSC Adv, 2024. 14(3): p. 2016-2026.

Zhou, X., et al., Poly(Glutamic Acid-Lysine) Hydrogels with Alternating Sequence Resist the Foreign Body Response in Rodents and Non-Human Primates. Adv Sci (Weinh), 2024. 11(16): p. e2308077.

Zuev, Y.F., et al., Water as a Structural Marker in Gelatin Hydrogels with Different Cross-Linking Nature. Int J Mol Sci, 2024. 25(21).

Li, P., et al., Simple Preparation and Bone Regeneration Effects of Poly(vinyl alcohol)-Resveratrol Self-Cross-Linked Hydrogels. ACS Omega, 2024. 9(50): p. 49043-49053.

Zhu, M., et al., Hydrogel-based microenvironment engineering of haematopoietic stem cells. Cell Mol Life Sci, 2023. 80(2): p. 49.

Xu, D., et al., Depth profiling via nanoindentation for characterisation of the elastic modulus and hydraulic properties of thin hydrogel layers. J Mech Behav Biomed Mater, 2023. 148: p. 106195.

Wu, Y., et al., Multiscale design of stiffening and ROS scavenging hydrogels for the augmentation of mandibular bone regeneration. Bioact Mater, 2023. 20: p. 111-125.

Wang, Z., et al., Bioactive prosthesis interface compositing variable-stiffness hydrogels regulates stem cells fates to facilitate osseointegration through mechanotransduction. Int J Biol Macromol, 2024. 259(Pt 2): p. 129073.

Xue, J., et al., Hydrogel Composite Magnetic Scaffolds: Toward Cell-Free In Situ Bone Tissue Engineering. ACS Appl Bio Mater, 2024. 7(1): p. 168-181.

Arpornmaeklong, P., et al., Effects of Calcium Carbonate Microcapsules and Nanohydroxyapatite on Properties of Thermosensitive Chitosan/Collagen Hydrogels. Polymers (Basel), 2023. 15(2).

Abudukelimu, K., et al., Preliminary study on the preparation of antler powder/chitosan/β-glycerophosphate sodium/polyvinyl alcohol porous hydrogel scaffolds and their osteogenic effects. Front Bioeng Biotechnol, 2024. 12: p. 1421718.

Zhu, Y., et al., Multifunctional injectable hydrogel system as a mild photothermal-assisted therapeutic platform for programmed regulation of inflammation and osteo-microenvironment for enhanced healing of diabetic bone defects in situ. Theranostics, 2024. 14(18): p. 7140-7198.

Wang, X., et al., Recent Strategies and Advances in Hydrogel-Based Delivery Platforms for Bone Regeneration. Nanomicro Lett, 2024. 17(1): p. 73.

Zhu, W., et al., Fabrication of Naturally Derived Double-Network Hydrogels With a Sustained Aspirin Release System for Facilitating Bone Regeneration. Front Chem, 2022. 10: p. 874985.

Zhu, L., et al., Flexible Fabrication and Hybridization of Bioactive Hydrogels with Robust Osteogenic Potency. Pharmaceutics, 2023. 15(10).

Zheng, L., et al., Oscillating Fluid Flow Activated Osteocyte Lysate-Based Hydrogel for Regulating Osteoblast/Osteoclast Homeostasis to Enhance Bone Repair. Adv Sci (Weinh), 2023. 10(15): p. e2204592.

Zhang, W., et al., Construction of artificial periosteum with methacrylamide gelatin hydrogel-wharton's jelly based on stem cell recruitment and its application in bone tissue engineering. Mater Today Bio, 2023. 18: p. 100528.

Yu, C., et al., GelMA hydrogels reinforced by PCL@GelMA nanofibers and bioactive glass induce bone regeneration in critical size cranial defects. J Nanobiotechnology, 2024. 22(1): p. 696.

Zhang, X., et al., A Bioactive Gelatin-Methacrylate Incorporating Magnesium Phosphate Cement for Bone Regeneration. Biomedicines, 2024. 12(1).

Yuan, S., et al., Immunoregulation in Skull Defect Repair with a Smart Hydrogel Loaded with Mesoporous Bioactive Glasses. Biomater Res, 2024. 28: p. 0074.

Yu, T., et al., An injectable and self-healing hydrogel with dual physical crosslinking for in-situ bone formation. Mater Today Bio, 2023. 19: p. 100558.

Neumann, M., et al., Stimuli-Responsive Hydrogels: The Dynamic Smart Biomaterials of Tomorrow. Macromolecules, 2023. 56(21): p. 8377-8392.

Round 2

Reviewer 1 Report (New Reviewer)

Comments and Suggestions for Authors

The addition of fresh sections and references to the paper has significantly enhanced the contribution. Tables 1-4 are also improvements since they provide a clear picture of material and biomodel categories based on their primary characteristics.

This manuscript is a resubmission of an earlier submission. The following is a list of the peer review reports and author responses from that submission.

Round 1

Reviewer 1 Report

Comments and Suggestions for Authors

The manuscript required major revisions as given below.,

1. Authors claim that "Bone degenerative diseases affect over 1.7 billion people globally". But the reference article for this sentence was published in 2018. Kindly include the current status with recent publications.

2. Authors must rewrite the entire manuscript as most of the references are very old (before 2019). For example., ref 1-3, 6, 8, 11, 12, etc. Authors can include recent publications (after 2022).

3. A detailed description about biomaterials must be included under the introduction section, as classification is included.

4. Although the title of manuscript is "Hydrogel-Based Scaffolds: Advancing Bone Regeneration through Tissue Engineering", there is no explanation about Hydrogel and their role in tissue engineering. This must be explained in detail under new-subheading.

5. Figure 1 explains the different properties of hydrogel. These traits must be explained along with "Hydrogel and their role in tissue engineering".

6.  There are numerous works on the usage of natural polymers for bone tissue engineering. But the authors have stated very few. For example., under the collagen section, only one in vitro experiment (published in 2013) is explained. Add at least 5 experiments for each polymer.

7. Why was  Silk and silk fibroin mentioned under the fiber section as they fall under the protein category?

8. All explanations under synthetic polymers must be elaborated.

9. Why "Growth factors and peptides are used to functionalize hydrogels?? This must be explained.

10. Entire manuscript must be checked thoroughly for grammatical mistakes and spellings. For example, the term "hydrogel" is misspelled as  "hidrogels" above the Growth Factors Section.

11. I am surprised to see only two figures for a review article. Include 7 more figure.

12. Future perspectives must be included.

13. Conclusion must be re-written.

14.  Since the authors have mentioned this sentence in abstract "The development of advanced hydrogel-based scaffolds holds great potential for revolutionizing bone tissue engineering and providing effective treatment options for patients with bone defects", a lot of clinical experiments must be included under natural and synthetic polymers.

Reviewer 2 Report

Comments and Suggestions for Authors

The review article « Hydrogel-based scaffolds : Advancing bone regeneration through tissue engineering” presents a catalogue of molecules used as building blocks for the production of hydrogels, and of growth factors and peptides used to stimulate bone regeneration.

General comments

1.       The authors do not explain why they do not present the cells which are commonly used in tissue engineering approaches.

2.       Several paragraphs dealing with specific compounds used to produce hydrogels do not even refer to articles describing the application of these hydrogels in bone tissue engineering. They often quote other reviews, similar to this one, which provide a similar catalogue of molecules. For example, they use reference 28 throughout the manuscript (quoted 25 times!!!). They should, instead, quote reviews which are focused on the particular compound (example: reviews on the use of gelatin in bone tissue engineering….). I also often found that a review was quoted instead of the original article to illustrate a specific data.

3.       Several quotations are completely inappropriate. To support a statement, the authors often quote a review article in which the sentence that they write in the manuscript is found in the middle of a paragraph, even though this article does not address at all the issue raised in the statement. I have indicated these references as “inappropriate” in the specific comments.

4.       There are many mistakes, wrong statements, and irrelevant paragraphs throughout the manuscript

Specific comments

Page 2

line 52. References 8 and 9 refer to very specific uses and properties of hydrogels and should not be used to illustrate the general purpose of this paragraph.

line 62-65. This sentence is not clear. Should all the matrices have the capacity to deliver cells? Many of the articles quoted in this review address the delivery of bioactive molecules, but do not include cells, so the . What do the authors mean by “the capacity to activate biomolecules”? Should the matrix activate endogenous molecules of the host? Or just deliver bioactive molecules? Please be more precise.

line 70. It is stated that there is a classification of scaffolds based on their consistency, but what are the different items/categories within this classification?

line 83. Quotation 21 is completely inappropriate.

line 85: references 18 and 23 are inappropriate.

lines 85-86: I do not understand this sentence.

line 90: reference 18 is not appropriate.

line 92: reference 24 is inappropriate

Page 3: The paragraph “bone degenerative diseases….a low resorption rate” should be moved after paragraph 2.1, as it addresses non-spontaneous bone repair and not bone physiology.

Lines 100-103. Are the 450 000 grafts really used to treat lesions caused by degenerative diseases?

Lines 104, 105: the terms “osteoconductive, osteoinductive and osteogenic” should be explicated here.

                Quotations 6, 7 and 28 are not appropriate.

Page 3, paragraph 2.1

                Line 121: neurogenesis is also a crucial step in bone repair and should be mentioned.

                References 2, 6, 13, 29, 30, 32 are inappropriate.

Page 3, bottom: the paragraph “hydrogel-based scaffolds for bone regeneration” should be number 2, not 1

Page 4

line 157. Reference 35 is inappropriate

I do not agree with the definitions given for osteoconductive, osteoinductive and osteogenic. Osteoconduction is a passive property and does not involve the activation of resident cells, but just their migration to the injury site. Osteoinduction indeed involves the activation of resident cells by exogenous factors. Finally, osteogenicity supposes the capacity to elicit bone formation in ectopic sites.

Page 5

Figure 2 : Panel A: replace osteoconduction with osteoconductive. Please replace “activate” with “recruits”; Panel B: What is BPM? ; Panel C: this figure seems to recapitulate the three different properties rather than focusing on the mechanisms of osteogenicity, this makes it difficult to understand.

Page 6

                Line 193: reference 36 is inappropriate

Line 200: the factors diffuse into the physiological fluids or interstitial fluid rather than dissolve in water.

Lines 221-222 and 229-230: “be readily degraded….and enzymes” is repeated.

Page 7

Collagen.

 Alkaline phosphatase activity and mineral deposition are function rather than viability parameters.

As this review addresses bone regeneration, it would be important to include a statement and references about the efficacy of collagen as a scaffold for bone tissue engineering. It would also be much appreciated if a comment about the drawbacks of collagen could be included.

Fibrin

I have the same comment as above: what are the limitations of fibrin?

Gelatin

Gelatin DOES NOT consist of a specific peptide sequence but of a mixture of peptides of different molecular weights.

As in the collagen and fibrin paragraphs and in most other paragraphs, reference 28 is used to support the properties of these molecules. Quoting a review which itself lists compounds used for tissue engineering does not provide any useful input. The authors should rather quote more focused reviews, for example in the case of gelatin, the review by Wu et al in Heliyon in 2024.

Lines 269-274: All the paragraph as well as reference 37 should be removed as they are addressing cartilage repair, not bone repair.

Page 8

Line 284: this is a paragraph on ECM properties, but the antimicrobial properties mentioned here are those of chitosan, so this does not illustrate the interest of ECM for bone tissue engineering.

Agarose

Line 296: I do not understand how the matrix structure conducts to cell-cell interactions.

This paragraph includes agarose-based hydrogels as useful scaffolds for bone tissue engineering without showing any data and reference supporting this notion!!! If this claim cannot documented, then the paragraph about agarose should be simply removed.

Alginate

Reference 28 is again quoted (twice !!!) and should definitely be replaced by a reference of a more focused review. Moreover, as for agarose, there is no example of an application of alginate in bone tissue engineering in this paragraph!! This points alginate as material for any biomedical application, not specifically bone.

Page 9

                Chitosan

Reference 28 is quoted 3 times in this paragraph!!! The authors could instead quote, for instance, Shan et al, Heliyon 2024, or any other one but not ref 28.

Page 10

                Dextran

This paragraph deals with the general properties and general potential applications of dextran but does not address applications in bone tissue engineering. I think this paragraph is useless as it is.

Silk

Once again, reference 28 is mentioned twice in this paragraph, with no specific review or original article quoted.

Silk fibroin

References about applications to bone repair should be included.

1.2 Synthetic polymers: the h is missing in the title

Reference 28 is back again, twice here. I was thinking that there is no need to write this review if everything is already in reference 28!

Page 11

                1.1.4

PLA DOES NOT FORM hydrogels, so this paragraph should be discarded

Again reference 28, twice! Instead, I would suggest Alavi et al, Artif. Organs, 2023 for a review focused on the use of PLA for bone tissue engineering.

1.1.5

PGA DOES NOT FORM hydrogels, so this paragraph should be discarded

Reference 28 is used again!

Reference 1 does not use a murine model but a rat model, and it is not PGA which is used but PLGA

1.1.6

Reference 28 once again

Line 463: Both references 54 and 55 are about human studies, not rabbit!

1.1.7

Reference 56 does not present any data on bone, whether trabecular or not. It is inappropriate here.

Page 12

1.1.8

Lines 489-492. This sentence mentions a specific study so the corresponding original article should be quoted instead of, again, reference 28.

Line 499 There is a mistake in the numbering of the paragraphs. This one should be 2

Lines 516-518. This sentence justifies the wide use of BMP growth factor by the use of an oligopeptide derived from this protein in one study, which does not really make sense.

Line 518. Reference 29 addresses PTH, not BMP!!!

Line 527. Reference 18 does not concern cancer cell proliferation. Please delete it

Page 13

Lines 531-524: The authors should quote the related original article, not the review.

Line 536: PDGF is not a potent source of anything, it is the platelets that are the source of PDGF. Please correct

Transforming growth factor-β1: TGF-beta stimulates the chondrogenic differentiation of MSC, not osteoblastic differentiation. This is stated in reference 40. So TGF beta is of no interest for bone tissue engineering and this paragraph should be removed.

Line 550: Reference 59 deals with bFGF, not TGFbeta

FGF, Line 561: please do not quote a review (reference 2) but the original article as the sentence describes a specific study. Same comment for line 569

Page 14

In the introduction to paragraph 1.2 the authors mention the effects of some peptides on macrophage polarization, but they should explain why this is relevant to bone repair. Also, reference 61 should be placed after “phenotypes”, line 581.

LL-37

There is absolutely no reference to an application for bone tissue engineering, so please delete this paragraph if you cannot include such application. Surprisingly, reference 68 indeed concerns the use of LL-37 to promote bone regeneration, but it is not mentioned in this paragraph!

PTHrP

Please quote a reference addressing the interest of PTHrP for bone tissue engineering.

Line 620: why is reference 68 placed here? It seems it doesn’t match with the subject treated here.

Round 2

Reviewer 1 Report

Comments and Suggestions for Authors

Authors have made all corrections as suggested. Manuscript can be accepted.

Author Response

Dear Reviewer, we consider that the corrections made improved the clarity and relevance of the manuscript, thanks for all your help.

Faithfully yours,

Maribel Aguilar-Medina, PhD.

FCQB-UAS

Reviewer 2 Report

Comments and Suggestions for Authors

The review article « Hydrogel-based scaffolds : Advancing bone regeneration through tissue engineering” presents a catalogue of molecules used as building blocks for the production of hydrogels, and of growth factors and peptides used to stimulate bone regeneration.

General comments

1.       The authors should explain why they present the growth factors and peptides, which can promote bone regeneration, but not the cells, which are commonly used in tissue engineering strategies.

2.       Several references are still completely inappropriate. To support a statement, the authors often copy-paste a sentence found in the abstract or introduction of an article, whose subject is absolutely not related to this statement. Alternatively, to illustrate a specific study, they quote a review instead of quoting the corresponding original article. Finally, in some instances, the referred article does not illustrate the statement.

3.       Although it is a review focused on bone tissue engineering (BTE) applications of hydrogels, some paragraphs such as PDGF and PTHrP lack references about applications of these factors to BTE.

4.       There are still many mistakes and wrong statements throughout the manuscript. Notably, sometimes a statement is attributed to a proposed reference but after checking it appears to differ from the real content of this reference.

5.       Some paragraphs should be entirely re-written.

6.       Some paragraphs address tissues other than bone and should be suppressed.

7.       Many paragraphs are not well-organized, with very specific uses of a compound is mixed with the description of its general properties.

Specific comments

Page 2

Graphical abstract: there are numerous spelling errors in the figure (elastic instead of elastic, PHRP instead of PTHrP, diferent instead of different, etc…. Please correct

Graphical abstract, legend, line 48. Please be more precise, as very few hydrogels have osteogenic properties, but they may acquire such properties by the addition of growth factors or osteogenic cells.

Page 3

Introduction: the sentence “Tissue engineering…renovation, replacement and tissue regeneration” in line 61-62 is repeated almost identically lines 81-83 (However…regenerating, replacing or repairing damaged or lost tissue). This repeat should be avoided. Moreover, each sentence is illustrated by different references. I do not understand the rationale of the choice of different references to illustrate the same concept.

Lines 61-75: references 3, 4, 6, 9, 10 are inappropriate. For instance reference 7 does not address the use of cells in TE, but the use of growth factors. References 9 and 10 refer to specific articles focused on the use of doped hydrogels, they do not address the issue of degenerative diseases.

Line 83. Please end the sentence with a mark (after [13-17]) and start a new one.

Line 89. Replace “materials” with “elements”.

Line 90. Whereas local cell populations are required for bone regeneration, exogenous cells must sometimes be included into the biomaterial to obtain osteogenesis.

Line 85. Reference 18 is inappropriate

Line 97. What deos “activate bioactive molecules” mean?

Line 104. References 10 and 25 are completely inappropriate.

Line 112. Some polymers used to produce hydrogels have positive charges (for example chitosan). Please modulate your statement.

Line 112. References 7, 24, 28, 29 are inappropriate.

Page 4

Line 114-115. The title of the manuscript mentions Hydrogel-based scaffolds, the sentence “natural and synthetic polymer scaffolds” suggests that the review will also consider non hydrogel scaffolds?

Line 119. Biocompatibility is not due to the presence of bioactive molecules. Moreover, most of the quoted natural polymers do not contain any bioactive molecules.

Line 120. Reference 30 concerns chitosan/gelatin hydrogels, not ECM.

Paragraph between lines 121 and 144 is mis-placed. It describes specific applications of different compounds used to generate biomaterials, so these examples should be inserted in the presentation of the different types of hydrogels, starting page 11. The introduction should focus on general properties.

Lines 121-122. The ECM is mainly the result of collagen deposition by cells, it is not a support for collagen.

Line 124. What does “compatibility with other molecules” mean? It is not clear.

Line 127. “modifications made during its synthesis”. I do not understand what this statement refers to. Do the authors mean that collagen contains modified amino-acids and that this may affect its enzymatic degradation? Please clarify.

Line 132-133. What is the relationship between the solubility of PEG and its property as a modifier of starch/PLA composite?

Line 147. The reference is not appropriate ; What does “dental organs” mean?

Page 6

The “bone healing” paragraph is not well-organized. It should start with spontaneous, physiologic repair of small fractures, followed by addressing large and critical-size defects.

Lines 173-186. This paragraph is a mess. It contains informations and statistics about transplants, bone void filler treatments, degenerative bone diseases, all these being mixed without any obvious logics.

References 15, 45 and 46 are inappropriate.

Line 198. The sentence “hydrogels based on ECM, collagen and/or gelatin have their molecules such as proteoglycans ….” Is meaningless. Collagen and gelatin do not contain proteoglycans!

Figure 1. panel B. replace “osteoinduction” with “osteoinductive” ; Why are cells like osteoblasts, macrophages and platelets mentioned here? There is no need to mention the origin of these factors, since they are to be used as additives in biomaterials

Page 8.

As discussed above, the paragraph between lines 227 and 246 should be placed at the beginning of section 2.1

References 50, 52 and 53 are inappropriate.

References 21 and 55 are inappropriate. Reference 56 is inappropriate: the sentence which refers to it is a perfect copy-paste of a sentence found in the beginning of the introduction of the article. Instead, the authors should quote the article which is quoted in reference 56, to illustrate their statement in lines 243-245.

Page 9

There is no reference illustrating the involvement of EGF, PDGF, TGFbeta in the activation of the immune system. Please provide some.

References 19 and 57 are completely unrelated to the subject of this paragraph.

Line 258. Please add a mark and start a new sentence.

Line 263. What does “and membrane” mean? Which membrane are the authors referring to?

Paragraphs 3 and 4 are very similar and should be grouped as a single one. Both describe general properties of hydrogels.

Line 266. The sentence “resembling the extracellular matrix and the repair…” is meaningless.

Line 273. I do not understand how a hydrogel can provide a nutrient environment for endogenous cells. This is very unclear.

Line 280. As written above, not all polymers have negative charges. For instance chitosan is positively charged, and starch and cellulose are neutral.

Lines 281-282. It is not because of water retention that polymers create a three-dimensional structural network, but because of the weak interactions between the polymer chains.

Figure 3. The property of “degradation” associated with synthetic polymers, should rather be attributed to natural polymers ; what does “solubilizing capacity” mean?

Figure 4 is complex and provides inconsistent information. The description of each compound concerns either mechanical (for example liquid consistency, resistance and elasticity) biochemical (controlled enzymatic degradation) or biological (upregulate osteopontin…) properties which are not comparable and hence do not allow for comparison of these compounds ; What does capacity carrier MBP mean?

Legend of figure 4. The sentence “despite…processes”, lines322-323 is absolutely meaningless.

Line 324. The authors certainly mean “osteoconductive”, not conductive. Please correct.

Line 324. Not all the hydrogels are osteoconductive, for example alginate is not osteoconductive.

Line 343. Most of the protocols used to purify collagen use acetic acid or urea-based solubilization. If there is a protocol that uses enzymatic treatment, please provide a reference.

Page 12

Line 347. The sentence “accounting for approximately 25% of the body’s total protein content” should not be placed here, where it does not mean anything.

The small paragraph lines 348-351 is useless as it describes an application in skin repair.

Line 364. Reference 4 describes a rat model, not patients. So this reference cannot be used to illustrate this statement.

Line 376 please replace “develop” with “development”.

Line 383. What does “this can result in a broad design space” mean?

Lines 387-390. What does “responsiveness to external stimuli” mean? Please explain, be more precise. “poor mechanical properties” does not follow correctly the previous part of the sentence. Please re-write carefully ; “Its applicability in cell culture”: Fibrin is not aimed at cell culture applications!!! ; “found within blood clot” does not connect correctly with the sentence, please re-write carefully.

Lines 392-396. This small paragraph is of no interest to bone tissue engineering and should be suppressed.

Page 13

Line 409. Please introduce a mark and start a new sentence.

Lines 409-413. This example is about cartilage, not bone. Please suppress.

Line 416 and 435. Reference 66 is in appropriate, as it concerns polysaccharides, not gelatin.

Line 439. Reference 79 is not relevant.

Line 442. Reference 31 is not relevant.

Line 445. “that collectively regulate pluripotency, proliferation and differentiation”. Of what?

Page 14

Line 456. After [82], please start a real phrase.

Lines 474-476. All this paragraph concerns cartilage and should be suppressed.

Page 15

Line 517. References 26 and 37 concern silk fibroi and chitosan, respectively, not hyaluronic acid.

Line 519. Lubrication and water retention are cellular responses???

Lines 528-530. This paragraph should be placed after line 520 as a general description of HA gels.

Line 537. In reference 94, it is collagen, not gelatin, which is used.

Page 16

Line 553. Instead of “adhesive properties”, do you mean cell; adhesive properties? Or something else?

Line 563-565. This example should not be placed in the middle of the general description of the chitosan properties. Moreover, the quoted article absolutely does not show any effect of chitosan on MSCs.

Lines 587-590. This example of an application in BTE should be placed after the general description.

Line 589. Please put a mark and start a new phrase.

Reference 105 is about cartilage, not bone. Please suppress.

Page 17

Line 605. In fact dextran by itself does not have any capacity to generate apatite crystals.

Page 20

Line 630. What does “direct phenotype” mean?

Line 648. Please put a mark and start a new phrase.

Line 669. Nothing in reference 44 shows that PFF is a viable substitute for trabecular bone tissue. This statement is not relevant.

Page 21

Line 676 Please put a mark and start a new phrase.

Line 680-684. The content of reference 123 is mis-interpreted here. Drug delivery is not mediated by PFF, but by a hydrogel which is included in a PFF scaffold. So this is absolutely not an evidence for drug delivery properties of PFF scaffolds.

Paragraph 4.2.3. It should be mentioned somewhere that PCL also has drawbacks such as its hydrophobicity ; High stability and cost-effectiveness” are repeated lines 692 and 695.

Line 696. PCL by itself does not have any tissue repair capabilities; it must be modified or supplemented. This is an information that should be highlighted ; Reference 125 does not show any tissue repair and should not be used here.

Line 706. Please insert reference 10 after “osteogenic properties”, not in line 709.

Line 705. None of the references quoted here mention the use of ECM. Please modify the paragraph.

Line 709. How is it possible to demonstrate biomineralization through alkaline phosphatase activity assays? The sentence would make sense after suppressing “assays”. Also, this article shows alizarin red staining, which is an evidence of mbiomineralization.

Lines 711-715. The release of BMP-2 shown in reference 125 is not achieved from PCL but from nanocapsules made with another compound. So this article does not show any ability of PCL as a drug delivery system.

Page 23

Line 727. In reference 131 BMSCs are not used.

Line 747. Why quote only one reference for BMP-2? References 124, 125 and 131 should also be included here.

Line 757-760. Reference 132, which quotes a manuscript published in 1965, does not have any relationship with the subject of this sentence.

Line 765. The quoted article does not study PDGF but platelet extracts.

Lines 766-767. Biominerals such as calcium and sodium? Calcium and sodium are salts, not minerals.

Comment on paragraph 5.1.2 Finally there is no valid reference for the use of PDGF for bone regeneration. However I found quite a few in Pubmed…

Page 24

Line 787. A more specific reference should be shown.

Lines 796-800. The interpretation of this article is confusing. It shows that genetically modified BMSC promote bone defect healing, and that in vitro, these cells overexpress FGF-2, which stimulates MSC differentiation towards osteoblasts. But there is no direct evidence that FGF-2 is responsible for increased bone formation.

Line 806. I could not find the information about M1 and M2 macrophages in reference 134.

Line 823. What are phosphate complexes?

Page 25

Lines 829-833. The lengthy description of the antimicrobial properties of LL37 is not useful as it does not concern bone applications. Instead, there some interesting articles that focus on LL37 and bone repair, which are missing here.

Paragraph 5.2.4. There is absolutely no reference supporting the use of PTHrP in BTE applications. Consequently this paragraph should be removed.

Line 864, line 902. Reference 149 is inappropriate.

Page 26

Line 916. I am not convinced that bioprinting technologies provide the necessary rigidity to biomaterials, unless a reference convinces me.

Author Response

Dear Reviewer, we received your comments, and carefully modified the manuscript to respond to your recommendations. Attached you will find the point-by-point answers.

Therefore, after these constructive revisions, we hope you will find the paper now suitable for publication.

Faithfully yours,

Maribel Aguilar-Medina, PhD.

FCQB-UAS
